

# Overview: Precipitation Characteristics and Sensitivities to the Environmental Conditions during GoAmazon2014/5 and ACRIDICON-CHUVA

Luiz A. T. Machado[1], Alan J. P. Calheiros[1], Thiago Biscaro[1], Scott Giangrande[2], Maria A. F. Silva Dias[3], Micael A. Cecchini[3], Rachel Albrecht[3], Meinrat O. Andreae[4,14], Wagner F. Araujo[1], Paulo Artaxo[5], Stephan Bormann[4], Ramon Braga[1], Casey Burleyson[6], Cristiano W. Eichholz[1], Jiwen Fan[6], Zheng Feng[6], Gilberto F. Fisch[8], Michael P. Jensen[2], Scot T. Martin[7], Ulrich Pöschl[4], Christopher Pöhlker[4], Mira L. Pöhlker[4] Jean-François Ribaud[1], Daniel Rosenfeld[9], Jaci M. B. Saraiva[10], Courtney Schumacher[11], Ryan Thalman[12] David Walter[4] and Manfred Wendisch[13]

[1]National Institute for Space Research (INPE), Sao José dos Campos, Brazil
[2]Brookhaven National Laboratory, Upton, New York, USA
[3]Institute of Astronomy, Geophysics, and Atmospheric Sciences, University of São Paulo, Brazil
[4]Max Planck Institute for Chemistry, Mainz, Germany
[5]Institute of Physics, University of São Paulo, São Paulo, Brazil
[6]Pacific Northwest National Laboratory, Richland, WA
[7]Harvard University, Cambridge, Massachusetts, USA
[8]Department of Aerospace Science and technology
[9]Hebrew University of Jerusalem, Israel
[10]Amazon Protection System (SIPAM), Manaus, Brazil
[11]Texas A&M University, College Station, Texas, USA
[12]Snow College, USA
[13]Leipzig Institute for Meteorology, Leipzig University, Leipzig, Germany
[14]Scripps Institution of Oceanography, University of California San Diego, CA 92037, USA

*Correspondence to*: Luiz A. T. Machado (luiz.machado@inpe.br)

**Abstract.** This is study provides an overview of precipitation processes and their sensitivities to environmental conditions, in the Central Amazon Basin, during the GoAmazon2014/5 and ACRIDICON-CHUVA experiments. Taking advantage of the numerous measuring platforms and instruments systems operating during both campaigns sampling cloud structure and environmental conditions during 2014 and 2015, the rainfall variability among seasons, aerosol loading, land surface type, and topography have carefully been characterized. Differences between the wet and dry seasons were examined from a variety of different perspectives. The rain rate distribution, the total amount of rainfall, and the raindrop size distribution (the mean mass-weighted diameter) were quantified for the two seasons. The dry season has a higher average rain rate than the wet season and reflects more intense rain. While the cumulative wet season rainfall amount was four times larger than the total dry season rainfall, reflecting in large total rainfall accumulation. The typical size and life cycle of the Amazon cloud clusters (observed





by satellite) and rain cells (observed by radar) were examined, as well their differences among the seasons. Moreover, we analyse the monthly mean thermodynamical and dynamical variables, measured by radiosondes to elucidate the differences in rainfall characteristics during the wet and dry seasons. The sensitivity of rainfall to the atmospheric aerosol loading is discussed with regard to the mean mass-weighted diameter and rain rate. This topic was evaluated during the wet season only due to the

5 insignificant statistics of rainfall events for different ranges of aerosol loadings and the low frequency of precipitation events during the dry season. The aerosol impacts on the cloud droplet diameter is different for small and large drops. For the wet season, we observe no dependence on land surface type on the rain rate. However, during the dry season, urban areas exhibit the largest rain rate tail distribution, and deforested regions have the lowest mean rain rate. Airplane measurements were performed to characterize and contrast cloud microphysical properties and processes over forested and deforested regions. The

10 vertical motion turned out to be uncorrelated with cloud droplet sizes, but the cloud droplets number concentration revealed a linear relationship to the vertical motion. Clouds over forest exhibit larger droplets than clouds over pastures at all cloud levels. Finally, the connections between topography and rain rate were evaluated, showing a higher rain rate over higher elevations for the dry season.

**1.      Introduction**

**1.1 - The Amazon Forest Climate**

The Amazon Forest is a huge area spanning more than 3,000 km in the east-west direction and approximately 2,000 km from north to south. The equator crosses the region, which is primarily located in the Southern Hemisphere and encompasses equatorial and tropical climates. Additionally, the northern part of the Amazon basin is influenced by the tropical Atlantic

Ocean, while the western edge is dominated by the Andes Mountains which rise more than 4,000 m above sea -level in the tropical and equatorial regions.

Cavalcanti et al. (2009) presented a detailed picture of weather and climate in Brazil, particularly in the Amazon. The dominant large-scale feature in the Amazon is the lack of major temperature gradients and the absence of baroclinic weather systems. However, this does not mean that there is a lack of convective organization. The main synoptic systems that approach the

25 region and alter the weather conditions are: a) the Intertropical Convergence Zone, mostly affecting the northern half of the Amazon; b) easterly waves coming from the tropical Atlantic (Diedhiou et al., 2010); c) upper tropospheric cyclonic vortices originating on the eastern coast of northeast Brazil and the associated upper air Bolivian High (Silva Dias et al., 1983 and Kousky and Gan, 1981); d) the South Atlantic Convergence Zone affecting the southern half of the Amazon, which has a major effect on the Amazon convective activity as a whole (Rickenbach et al., 2002); and e) the northward propagation of convective

clouds (Siqueira and Machado, 2004) and the remnants of mid-latitude cold frontal systems that may propagate northward, sometimes beyond the equator, resulting so called "friagem"events (Marengo et al., 1997). Within the basin, convection is often organized into squall lines (Cohen et al., 1995), frequently occurring as large systems originating at the northern coast triggered by local sea breeze circulation (Grecco et al., 1995). Some of these squall lines propagate to central Amazonas, dissipating during the night and reactivating the next day by diurnal heating.



Climate controls on the Amazon basin rainfall come from episodes of El Niño/La Niña, defined by the tropical Pacific Ocean sea surface temperatures (SSTs) and from the tropical Atlantic SSTs (Marengo et al. 2013, 2016). Warm tropical Atlantic SSTs are associated with drought conditions in the Amazon region. During El Niño episodes, most of the Amazon basin experiences below average rainfall, while La Niña cases are associated with above normal rainfall. The convective activity in most of the

5 Amazon basin is part of the South American monsoon system (SAMS), which is associated with distinct wet and dry seasons (Silva Dias and Carvalho, 2016).

Horel et al. (1989) used satellite downward longwave radiation to characterized the seasonal variation in the Amazon region, with typical wet and dry seasons including two transition periods. Machado et al. (2004) defined regionally, inside the Amazon basin, the driest month and the dry season duration. The dry season duration varies from only one month in the northwest

sector to 3-4 months in southeastern Amazonas, For the Central Amazonas region, July and August are typically the driest months. Convection in Amazonas is more intense during the dry to wet transition season when thunderstorms have more lighting activity (Albrecht et al., 2011) and when they are more sensitive to aerosol loading and topography (Gonçalves and Machado, 2014). During the transition from the dry to the wet season is influenced by complex interactions between smoke-derived aerosols and deep convective clouds occurs (Albrecht et al., 2011). Although the average Convective Available

Potential Energy (CAPE) seasonal variability is small, the tail of the CAPE seasonal distribution exhibit relatively higher values during dry to wet season transition than during the wet season (Williams et al., 2002). During the dry season, the aerosols produced by biomass burning in central South America impact a larger area, reaching the tropical Pacific, subtropical South America and South Atlantic (Andreae et al. 2001; Freitas et al., 2005, 2016, Camponogara et al., 2014).

While Amazonas region exhibits strong seasonal variations of atmospheric circulation and related precipitation pattern, the

20 diurnal cycle is typically the same throughout the year. Most of the region has an afternoon peak of convective activity; however, there are selected areas where quite intense nocturnal systems are observed and where seasonality is more pronounced (Saraiva et al., 2016). The diurnal cycle of convection has a strong link to the underlying surface: its topography (Laurent et al. 2002), deforestation (Saad et al. 2010) and large rivers (Santos et al., 2014, Silva Dias et al. 2004), demonstrating a link to surface features (Machado et al., 2004, Silva Dias et al., 2002). Additionally, large rivers impact rainfall evolution

through the convergence of the river breeze with ambient flow (Fitzjarrald et al. 2008). Adams et al. (2016) showed that one of the climate models central problem related to Amazon diurnal variability of convection and rainfall is the transition from shallow to deep convection, which occurs on a time scale of approximately three hours.

The evolution of the boundary layer in the Amazon region has been studied, during intensive field observations conducted in different sub-regions in the Amazon Basin: The Amazon Boundary Layer Experiment (ABLE 2A, 2B, see Harris et al. 1988,

1990), the Anglo-Brazilian Amazonian Climate Observation Study (ABRACOS, see Gash et al. 1996), the Large-Scale Biosphere-Atmosphere experiment in Amazonia (LBA, see Silva Dias et al., 2002), the Cloud Processes of the Main Precipitation Systems in Brazil: A Contribution to Cloud-Resolving Modelling and to the Global Precipitation Measurement (CHUVA, Machado et al., 2014) combined with ACRIDICON (Aerosol, Cloud, Precipitation, and Radiation Interactions and Dynamics of Convective Cloud Systems, Wendisch et al., 2017), and the Green Ocean Amazon GoAmazon2014/5 (Martin et





al., 2017). Fisch et al. (2004) indicated that the evolution of the boundary layer in the Amazon is linked to land cover and soil moisture, with a deeper mixed layer in the dry season over deforested areas and a shallower mixed layer over forest. During the wet season, there are small differences between the mixed layer evolution over forested and deforested regions.

During the dry season, the lower atmosphere is polluted by high aerosol concentrations caused by both, biomass burning and longer aerosol lifetime because of reduced precipitation (Artaxo et al., 2002 and Martin et al., 2010). During the wet season, the atmosphere is mostly clean and convective, and the landscape is referred to as the Green Ocean (Roberts et al., 2001, Williams et al., 2002 and Andreae et al., 2004) because convection resembles storms over blue oceans, where the warm phase in clouds generally produces rain. Large urban areas, however, introduce perturbations into the pristine air (Martin et al. 2016, 2017).

The complex physico-chemical interaction observed in the Amazon basin includes the processes of rainfall formation, diurnal, seasonal, inter-annual cycles, cloud spatial organization, the mechanisms controlling Cloud Condesantion Nuclei (CCN), the interaction between the vegetation, atmospheric boundary layer, clouds and the upper troposphere. These processes are all in perfect combination, resulting in a stable equilibrium climate that produces rainfall equivalent to 2.3 metres along the 6.1 million square kilometres of the Amazonas basin, equivalent to 27 trillion metric tons of rain each year on average. However, this amazing, complex mechanism can be modified by human activities. Recent results prove and quantify (Fu et al., 2013) how this stable environment can be disturbed and promptly move to another point of equilibrium far from the one that produces abundant fresh water, keeps the forest alive and has a main role in controlling the global atmosphere circulation and energy distribution.

## 1.2 Knowledge about Cloud Process in the Amazon Acquired During Field Campaigns

In particular, the most recent GoAmazon2014/5, and CHUVA-ACRIDICON measurement campaigns provided Amazonas with a comprehensive dataset to elucidate the complex aerosol-cloud-precipitation interaction. The GoAmazon observations, collected over two years, have delivered a wealth of data to study aerosol-cloud-precipitation (ACP) interactions (Martin et al., 2016). During the two intensive operation periods (IOPs), conducted during the wet and dry seasons, additional airplane data were collected in the IARA (Intensive Airborne Research in Amazonas); Martin et al., (2017) and the ACRIDICON campaign (Wendisch et al., 2017). The overall data collected in the framework of GoAmazon also includes the CHUVA project (Machado et al., 2015) and several other initiatives, which compiled the most complete dataset in Amazonas to study the atmospheric chemical and physical interactions. GoAmazon2014/5 were held in the environs of Manaus city, the capital of Amazonas State. Manaus is a city of around two million population, in the middle of Central Amazonas, serving as a natural laboratory to explore the urban pollution effects on the Amazonas background environment.

Recent work by Gerken et al. (2015) showed a strong enhancement of ozone concentration close to the surface during storm downdrafts in the central Amazon and discusses the important effect of storm downdrafts bringing higher ozone concentrations from middle-higher altitudes. The same effect was found by Betts et al. (2002) for the southwest Amazon during LBA. Wang





et al. (2016), using data from the airplane (G1), describe the mechanism for maintaining the aerosol concentration in the pristine Amazonian boundary layer. The aerosol loss by precipitation scavenging at the surface is replaced by the storm downdraft fluxes that bring a high concentration of nano-sized particles from the upper atmosphere during precipitation events. These nanoparticles combine with the oxidation products of VOCs (Volatile Organic Compounds) to form CCN at the surface,

assist the formation of clouds. Measurements by the G1 and by HALO (High Altitude and Long-Range Research Aircraft) show a very high concentration of nanoparticles in the upper troposphere, with concentrations up to 65,000 particles per cm3 (Andreae et al., 2017).

Aerosols particles influence cloud formation. Cecchini et al. (2016) highlight the effects of the Manaus aerosol pollution plume on the cloud droplet size distribution during the wet season when only a small sensitivity would be expected. They described

the significant influence of the Manaus pollution plume in reducing the size and increasing the number of cloud droplets as well as the total liquid water content. The ACRIDICON-CHUVA campaign collected in-situ data during 14 research flights with the HALO research aircraft (Wendisch et al., 2016). The high numbers of flight hours inside growing cumulus clouds allowed a sensitivity analysis of the aerosol concentration and of the thermodynamic effects in the warm phase of cumulus clouds. Cecchini et al. (2017a) demonstrated that an increase of 100 % in the aerosol concentration led to an 84 % increase in

the droplet number concentration, but the same relative increase in vertical velocity corresponded only to a 43 % change. Braga et al. (2017) compared HALO microphysical probe measurements of cloud droplet number concentration with a parameterization based on CCN and updraft at cloud base. Jäkel et al. (2017) presented a new methodology to retrieve the vertical distribution of hydrometeor phase using cloud-side reflected solar radiation measurements and discussed the mixed phase layer as a function of aerosol loading. Giangrande et al. (2016) presented the statistical behaviour of cloud vertical

motions as function of season, instability and convective inhibition. Burleyson et al. (2016) discussed the diurnal cycle and spatial variability of deep convection among the different GoAmazon sites. Giangrande et al. (2017) presented an overview of cloud, thermodynamics, and radiation interactions.

Preceding GoAmazon2014/5, ABLE-2 and LBA collected cloud and rainfall data used to understand the rainfall variability and its interaction with surface vegetation, topography and aerosols. The ABLE-2 project consisted of two expeditions: the

first in the Amazonian dry season (ABLE-2A) during July-August 1985 and the second, in the wet season (ABLE-2B) during April-May 1987 (Harriss et al. 1988 and 1990). Grecco et al. (1990) described the rainfall and kinematics of the Central Amazona using GOES (Geostationary Operational Environmental Satellite) imagery, revealing the importance of the tropical squall lines on the rainfall regime of the Amazonas. Some years later, Garstang et al. (1994), Grecco et al. (1994) and Cohen et al. (1995) provided a detailed description of the tropical squall lines. The TRMM-LBA campaign was designed to calibrate

the TRMM (Tropical Rainfall Measuring Mission) satellite. The observations were conducted in southern Amazonas, the arc of deforestation, during the wet season. Several studies contributed to the understanding of the rainfall variability at different scales. Machado et al. (2002) discussed the complex diurnal cycle interaction at a synpotic scale, Laurent et al. (2002) examined the mesoscale convective system initiation and propagation, and Rickenbach et al. (2004) showed the importance of the nocturnal clouds in the southwest Amazonas rainfall. Silva Dias et al. (2002), Petersen et al. (2001), and Cifelli et al. (2002)



are some of the studies published using TRMM-LBA data to describe the microphysical properties of the rainfall field, the cloud processes, and the biosphere interactions. In addition, different rainfall features were detected associated with wind regimes, easterlies and westerlies in southern Amazon associated to break and active phases of the South American Monsoon System (Silva Dias and Carvalho, 2016, Rickenbach et al 2002). In Northwestern Amazon, northerlies and southerlies are associated with more stratiform and convective systems, respectively (Saraiva et al 2017).

There are other studies discussing the rainfall regime in Amazonas. For example, Tanaka et al. (2015) described the influence of the river and the city of Manaus in the diurnal cycle based on rain gauge data. Dos Santos et al (2014) used satellite rainfall products to define the features associated to the river breeze associated to the Negro, Solimões and Amazon rivers. Fitzgerald et al. (2008) described the Tapajos River effect on the rainfall and Silva Dias et al (2004) showed the wind structure that favours cloud formation on the upwind side of the Tapajós river during daytime. Negri et al. (2000) used passive microwave radiances to construct a 10-year climatology of the Amazonas rainfall. Saraiva et al. (2016) described the general statistics of the Amazonas rainfall using the meteorological S-band radar operational network and discussed the diurnal cycle as well the relationship between reflectivity and the cloud electrification process.

All these studies established the basic knowledge about the rainfall statistics and related processes in Amazonas, providing a new perspective of research in Amazonas and elucidating several aspects of the ACP interactions. The studies associated with field campaigns covered specific seasons (normally the wet season) or resulted from sparse rain gauge or indirect measurements with low space and time resolutions. In GoAmazon2014/5 the extensive set of rainfall data collected by the S, X, and W band radars, airplanes, disdrometers, vertical pointing radar, rain gauges, microwave radiometers, ceilometers, and LIDARs provides a comprehensive view of the main variabilities and characteristics of the precipitation in Central Amazona. Giangrande et al. (2017) present an overview covering the clouds aspects, mainly focusing on the diurnal cycle and the impact on the radiative effects and thermodynamics effects. This study presents an overview of the rainfall characteristics and sensitivities to vegetation, topography, and aerosols particles and evaluates the seasonal variability. The main goal is to discuss the sensitivities of the main processes controlling rainfall over Central Amazona, employing a relatively long time series (2014-2015) of data based on the comprehensive dataset collected during GoAmazon2014/5 and complemented by aircraft measurements made during ACRIDICON-CHUVA.

Section two describes the data and methodology employed in the study. Section 3 presents the results and discussions of the seasonal rainfall characteristics and sensitivities to aerosol, vegetation and topography, and Section 4 summarizes the major findings.

## 2.       Data and Methodology

Several instruments were employed in this study. This section describes the instruments and the data processing procedures.

A laser precipitation disdrometer (PARSIVEL, see Löffler-Mang and Joss, 2000) measures the size and terminal velocity of hydrometeors that pass through the detection area sampled by a laser beam (54 cm$^2$) in a specific time interval. Two different





PARSIVEL disdrometers were used during the whole campaign: from CHUVA Project from January to September 2014 and another from ARM (Atmospheric Radiation Measurement) from September 2014 to October 2015. Raindrops larger than 5 mm were eliminated as to best match the co-located rain gauge accumulated rainfall, and a complementary filter was applied, as described by Giangrande et al. (2016). The drop size distribution (DSD) and all the respective rain rates (RR, in mm h$^{-1}$)

and mean mass-weighted diameter ($D_m$, in mm) were obtained in 5-minute intervals for periods with RR $\geq$ 0.5 mm h$^{-1}$, as suggested by Tokay et al. (2013).

The Doppler radar S-band dataset consists of retrievals from the Manaus radar operated by the Amazon Protection System (SIPAM). The reflectivity and rain rate fields were computed using the 2.5 km SIPAM Manaus S-band Constant Altitude Plan Position Indicator (CAPPI) for each radar volume, every 10 minutes. The corrected radar reflectivity from each volume were

10 interpolated to a fixed grid on which the rainfall products were generated. Specific procedures were applied to the dataset to compute rain rates from reflectivity, to reduce noise, and to improve data quality. First, rain rates were computed using a Z-R relationship adjusted to the region using 2014 wet-season impact disdrometer data: $Z=174.8R^{1.56}$. The maximum and minimum rain rate considered were 160 mm h$^{-1}$ and 0.2 mm h$^{-1}$. respectively. Rain rate was not computed when the radar beam had less than 10% quality reflectivity values (non-null reflectivity). Finally, a range filter was applied to remove the pixels closer than

15 10 km and farther than 135 km from the radar.

The Doppler radar X-band dual polarization dataset was obtained by the mobile Meteor 50DX Selex radar during the CHUVA Project (Schneebeli et al., 2012). The radar data underwent three main processes steps including differential phase shift (PhiDP) filtering and specific differential phase (KDP) derivation, differential reflectivity (ZDR) offset correction, and horizontal reflectivity (Zh) and ZDR attenuation correction. The uncorrected raw differential phase shift has a noisy signal that needs to

20 be filtered and smoothed before the calculation of its range derivative (KDP). Several methods can be used, such as a moving average, median filters, and linear programming approaches. In this study, we used the finite impulse response (FIR) filter, based on Hubbert and Bringi (1995). Once the filtered PhiDP profile is obtained, KDP is calculated using a least squares linear fit. To verify and calibrate the accuracy of the differential reflectivity measurement, a vertical pointing, rotating scan (also known as "bird-bath" scan) was incorporated into the X-band scan strategy. During light precipitation and in the absence of

25 strong winds, the vertical and horizontal returned signal of a vertically-oriented beam should be the same. Differences between the Horizontal-channels and Vertical-channels may appear due to poor calibration between the channels, random effects, beam-filling, or side-lobe clutter contamination, among other factors (Gorgucci et al., 1999). Although the standard calibration was performed, careful examination of the ZDR behaviour before and after these changes was necessary. We selected all of the observations with no radar gates higher than 30 dBZ below 2 km and analysed the overall ZDR values and the temporal

changes in the mean value. A persistent, positive ZDR offset (approximately 0.5 dB) were found and applied to the data. After these steps, we applied the attenuation correction. X-band radars are more prone to signal attenuation due to rain than C- and S-band radars. It is therefore mandatory to correct the signal for attenuation prior to any analysis using reflectivity data, if such a correction is possible (Schneebeli et al., 2012). With a dual-polarization system, one can use the differential phase shift to calculate the attenuation due to rain. We applied the ZPHI method (Testud et al., 2000) to the whole dataset where dual-





polarization moments were available. The corrected Zh and ZDR were then ready to be used as input to rainfall analysis or hydrometeor classification studies.

The cloud droplet size distributions were derived from the HALO measurements using a cloud droplet probe (CDP, Lance et al. 2010; Molleker et al. 2014 and Wendisch and Brenguier, 2013). This instrument measures the droplet size distribution

within the size range of 3 µm to 50 µm based on the hydrometeors forward scattering properties. The DSD is sorted into 15 size bins for each measurement cycle. The probe was operated at a 1 Hz frequency. The major sources of uncertainty of the instrument are as follows (Weigel et al., 2016): (a) uncertainty in the cross-section area ($0.278 \text{ mm}^2$ +/- 15%), (b) the relatively small sample volume (cross-sectional area multiplied by aircraft speed), and (c) counting statistics for each size bin. As noted by Molleker et al. (2014), the CDP uncertainty is approximately 10 %. Braga et al. (2017) performed an intercomparison

between the CDP and the other HALO cloud probes and concluded that they agree within instrumental uncertainties. The vertical wind component (w) was measured by the Basic HALO Measurement and Sensor System (BAHAMAS) located at the nose of the aircraft (Wendisch et al., 2016) and calibrated according to Mallaun et al. (2015). The uncertainty in w is approximately $0.3 \text{ m s}^{-1}$.

Using the disdrometer or the CDP, the mean mass-weighted diameter ($D_m$) was computed as the ratio between the fourth and

third moments (liquid water content) of the DSD (see Bringi et al., $D_m$ 2002 for a detailed description). For every five minutes of a continuous rainfall event (defined as RR $\geq$ 0.5 mm h$^{-1}$), the moments were computed using a parameterization by Tokay and Short (1996) parameterization.

A condensation particle counter (CPC, TSI 3010) in the Aerosol Observing system instruments from ARM measured the number concentration of aerosol particles at 10 m above ground level at the T3 site.(main GoAmazon site at Manacapuru, see

Martin et al., 2016 for a detailed description). The data was averaged to a five minutes time interval covering the period from January 2014 to March 2015 (Thalman et al., 2017). The background or polluted conditions were defined based on the specific CPC distribution for each season, by the threshold value associated with the 33% and 66% percentile respectively, for dry and wet season. The threshold values and details are presented in the specific section.

The total number concentration of cloud condensation nuclei particles ($N_{CCN}(S)$) was measured with a continuous-flow stream

wise thermal-gradient CCN counter (model CCN-200, DMT, Longmont, CO, USA) (Rose et al., 2008). The measured aerosol was sampled by the HALO aerosol submicrometer inlet (HASI). Particles with a critical supersaturation (S=0.52 $\pm$ 0.05 %) are activated and form water droplets. Water droplets with a diameter $\geq$ 1 µm are detected by an optical particle counter. Details about the measurement mode can be found by Andreae et al. 2017. The error in supersaturation resulted from the calibration uncertainty, as described by M. Pöhlker et al. (2016); it is estimated to be in the range of 10%.

This study considers the wet and dry seasons as the months of January to March and August to October respectively. Some instruments only operated during the two GoAmazon2014/5 IOPs: IOP1, during the wet season and IOP2 during the dry season, corresponding to February to March and 15 September to 15 October, respectively. The S-band radar data is available for both years (2014 and 2015), while the X-band only operated in the 2014 IOPs.





## 3.     Results and Discussion

This section first discusses the rainfall characteristics and variability by comparing the wet and dry seasons to establish the main differences. Section 3.2 discusses the studies of sensitivities to aerosols, vegetation and topography.

### 3.1     Rainfall Seasonal Variability

The seasonal variability will be presented from different perspectives including general patterns, regional differences, characteristics of the observations from satellite (clouds) and radar (rainfall), the drop size distribution, rainfall vertical profile, and hydrometeor populations.

### 3.1.1     General Precipitation and Thermodynamic Patterns.

Amazonas has a distinct seasonal variability with distinguished wet and dry seasons. The definitions of wet season length and beginning depends on the region within the Amazonas Basin. The mean monthly rainfall (2014-2015) for the wet season was 369 mm (considering a month of 30 days) and only 87 mm for the dry season. These numbers show the large difference expected between the two seasons. However, from the rain rate point of view (considering only when is raining), the seasonality also presents differences, the mean rain rate (computed in 5 minutes intervals) for the wet season was 7.7 mm h$^{-1}$ compared to 9.4 mm h$^{-1}$ for the dry season. Therefore, although the dry season has approximately 4 times less accumulated rainfall than the wet season, the average rainfall event has a higher rain rate. Figure 1 clearly reveals this feature, where the relative rain rate distribution for the wet season shows higher relative frequency for RR < 20 mm h$^{-1}$, than that of the dry season. On the other hand, the dry season shows higher relative frequency for RR> 20 mm h$^{-1}$. There was only one event during the wet seasons with record rain rate of approximately 100 mm h$^{-1}$ within a few minutes only. This is the reason why the wet season has a maximum rain rate which is higher than during the dry season, but the relative population of the dry season rainfall event is more pronounced towards high RR than during the wet season. This distinctive feature has important consequences for the cloud microphysical and macrophysical structures. The main reason for this difference is associated with the instability and cloud cover. Figure 2A shows the CAPE distribution for the wet and dry seasons in 2014. The dry season has a larger CAPE than the wet one, and the frequency of CAPE exceeding 2000 J kg$^{-1}$ is higher during the dry season. The wet season has typical monsoonal rainfall, with widespread moderate rain, in contrast to the more isolated and intense rainfall during the dry season. This characteristic of rainfall events with a higher rain rate during the dry season is explained based on the seasonal difference in the thermodynamic parameters. Comparing these seasons, important differences can be clearly noted in Fig. 2. The dry season has a larger CAPE, higher CINE (Convective Inhibition Energy), less available water vapor, a higher cloud base, and higher shear than the wet season. CAPE increases from March to September and the largest tail distributions occur at the end of the year when humidity increases and cloud base decreases. During the dry season, only regions with strong forcing can produce convective clouds to use the higher CAPE and shear available to produce organized convection. Gonçalves et al. (2014) show that the higher rain rate (radar reflectivity values larger than 35 dBZ) during the dry season mainly occurs over




the higher topography in Amazonas (section 3.2.1). The higher CIN and, smaller amount of water vapor reduces the occurrence of convection, but when convection can develop, it has all the ingredients to be deeper. Machado et al. (2004) explains the more intense convective clouds during the dry to wet season transition may result from less ''competition" of surface moisture convergence to feed cumulonimbus clouds because there is a smaller number of rain cells.

### 3.1.2    Cloud Clusters and Rain Cells-Size Distribution

Cloud clusters and rain cell calculations were computed using the GOES-13 satellite images and S-band radar employing the algorithm called Fortracc (Forecasting and Tracking Cloud Clusters see Vila et al., 2008). A cloud cluster is defined by
connected ensembles of pixels with brightness temperatures (BT), for channel 10.5 μm, colder than 235 K as defined by Machado et al. 1988. Rain cells is defined as connected ensembles of pixels in the radar 2.5 km CAPPI with reflectivity larger than 20 dBZ. Embedded in the cloud clusters, quite often rain cells are observed when clouds start to have raindrops. Considering the wet and dry seasons, the typical cloud cluster size and lifespan have 75-km effective radius (hereafter called as radius) and a 1.5-hour, respectively. The typical rain cell size has a 7.5-km radius and an 0.6-hour lifespan. On average,
cloud clusters are 10 times larger and have lifetimes approximately three times that of the rain cells. These are the average characteristics and, there is a wide range of cloud cluster sizes. Cloud clusters can have more than a 300-km radius and a lifetime longer than 24 hours, while rain cell sizes occur up to approximately 60-km radius and lifetime of a couple of hours. Figure 3 shows the dry and wet cloud cluster and rain cell size distributions and the differences among them. The basic size distribution is not very different between the two seasons because cloud cluster size distribution has a power law size
distribution, as shown by Machado et al. (1992), but some clear characteristics can be noted if the difference is computed. The wet season has more small and large rain cells and cloud clusters than the dry season. The dry season produces more rain cells in the range of a 10-km radius and cloud clusters of an approximately 40-km radius. The ratio between the cloud cluster and rain cell average radii during the wet season is much higher because of the larger stratiform cloud decks typical of a monsoon cloud regime. The thermodynamics of the dry season environment discussed in the preceding section favours the organization
of more compact and active convection, with more intense rain rates but with an accumulated rainfall 4 times smaller.

### 3.1.3    Rainfall Mean Mass-Weighted diameter for the Dry and Wet Seasons

An important aspect to be evaluated is related to the different cloud processes between the two Amazonian seasons. Are there
important microphysical differences between raindrops in the wet and dry seasons or are only the rain rate and rainfall frequency different? The way to investigate these features included the deployment of disdrometers and a dual-polarimetric radar. It is important to consider that this study focuses on rainfall and raindrops and that seasonal differences in cloud droplet size distributions may be quite different.





For instance, the effect of the aerosol concentration on the cloud droplets, in shallow convective clouds, generally reducing the size and increasing the concentration for a given liquid water content, is well known (Cecchini et al., 2016, among several other studies). However, if a polluted cloud transitions from shallow to deep convection, aerosols can invigorate clouds (Rosenfeld et al., 2008, Koren et al., 2012, Gonçalves et al., 2014). Giangrande et al., (2017) present the G1 airplane cloud

particle distribution measurements during GoAmazon2014/5, showing the predominance of larger cloud droplets, in warm clouds, in the wet season. The in-situ cloud droplet data were collected for a shallow cloud population. The result is very different when the comparison among seasons is made using disdrometers that measure raindrops at least 100 times larger in their mean mass-weighted diameter ($D_m$). As the raindrop diameter depends on the rain rate which is different between the two seasons, the comparison between $D_m$ for the dry and wet seasons were performed as a function of the rain rate in 5 mm h$^{-1}$

intervals. Another important feature to be evaluated in the comparison is the different frequencies of convective and stratiform clouds in the wet and dry seasons. As discussed by Giangrande et al. (2017), the wet season has higher occurrence of stratiform clouds than the dry season; hence, the $D_m$ evaluation should be performed separately for convective and stratiform clouds. The cloud classification employed in this study was computed using the radar wind profiler (RWP) and ancillary data as described by Giangrande et al. (2017) from March 2014 to December 2015. Clouds were classified based on the predominant cloud type

in the warm cloud layer. Convective clouds include strong and weak convection while stratiform clouds include the classification of the stratiform with a well-defined bright band and stratiform with a not well-defined bright band.

Figures 4a and 4b show the $D_m$ for the wet and dry seasons as function of the rain rate for convective and stratiform clouds. The arrows on the x-axis mark distributions where the averages are different, with a statistical significance of 95%. For convective clouds, it is clear that the mean mass-weighted diameter is larger during the dry season for a given rain rate. This

result suggests the different cloud processes generating larger rainfall drops, even if the shallow clouds have larger droplets in the wet season. The differences in shallow clouds during the dry season may be due to the reduced humidity, as shown in Fig. 2c, which can reduce supersaturation and increase droplet evaporation via entrainment. However, if the cloud evolves to become a deep convective stage, the higher cloud base, shear, and CAPE induce stronger vertical motions and mesoscale organization, generating higher amounts of ice (shown in next section) and forming larger raindrops through melting of large

ice particles such as snow and graupel. For stratiform clouds (Fig. 4b), the difference between the two seasons is much smaller and only significant for very small rain rates.

### 3.1.4 Cloud Vertical Profiles for the Dry and Wet Seasons

The results presented above discuss the rainfall on the ground. In order to comprehensively understand the cloud processes associated with the dry and wet seasons includes the evaluation of the hydrometeor vertical profiles of precipitating clouds. The X-band dual polarization radar was installed in the T3 site and operated during the two GoAmazon2014/5 IOPS in February-March and September-October 2014. To avoid wet radome attenuation effects and obtain good dual polarization measurements with good vertical coverage, the data analysed to build the next figures included only cases without rain over





the radar and for data collected within the 10 and 60-km radius range. Based in the radar strategy described in the data section, the volume scan was processed with attenuation correction and a ZDR offset to build the contoured frequency by altitude diagram (CFAD) for the dry and wet seasons for the reflectivity (Z), specific differential phase (KDP), differential reflectivity (ZDR), and the horizontal and vertical correlation coefficient that measures the consistency of the H and V returned power and

phase for each pulse.

Figure 5 shows the reflectivity CFAD for the dry and wet seasons. One can note the typical stratiform and convective pattern for the wet and dry seasons, respectively. For the wet season, the bright band is very clear with a pronounced peak of reflectivity at approximately at 4 km, which corresponds to the layer where ice is melting and increasing the reflectivity. Moreover, the upper levels above the melting layer have less intense reflectivity, demonstrating the less intense convective process in the

majority of cases during the wet season. For the dry season, the typical convective profile with a higher reflectivity in the lower levels corroborates the higher rain rates during this season. In addition, the mixed and glaciated layers have a more frequent occurrence of high reflectivity values, indicating the presence of large ice hydrometeors.

Figure 6 shows the CFAD for the dual polarization parameters, ZDR, KDP and co-polar correlation coefficient. The hydrometeor response to the transmitted signal depends on several factors that can change the characteristics of the measured

signal, such as hydrometeor orientation by the electrification field (see Mattos et al., 2017 for a detailed discussion), ice density, and different crystal shapes. Of course, there are also other possible effects that may impact the data quality, such as resonance and partial beam-filling. Although the CFADs are not completely different, as these parameters have a small range of variation and the limitations described above, some clear differences among the seasons can be observed. The ZDR that mostly describes the ice orientation has a larger frequency of near-zero or negative signals in the dry season than during the wet season. This

can be associated with the crystal orientation by electrical field, as the dry season has much more lighting than the wet season (see Williams et al. 2002) and possibly ice vertically-oriented shapes such as graupel. The KDP distribution shows considerably larger values in the warm layer, in the dry compared with wet season, indicating the higher rain rate and higher frequency of positive values in the mixed phase is probably associated with intense updrafts, as shown by Giangrande et al., (2016). The correlation coefficient highlights an interesting feature. The dry season has higher correlations at approximately 8

25    km, demonstrating a more homogenous ice content. It seems that there is a clearer distinction between the mixed phase and the glaciation phase above 8 km. The wet season correlation coefficient is more homogenous with height inside the cloud. Cecchini et al. (2017b) and Jakel et al. (2017) discuss the larger efficiency of clouds forming in a clean environment to produce ice. This feature can indicate that clouds forming during the wet season have a smaller mixed phase than the clouds forming during the dry season in a much more polluted environment.

### 3.2    Rainfall Sensitives to Aerosols, Topography, and Vegetation

This section studies the rainfall sensitivities to different surface types, topography and aerosol number concentrations. Ancillary data describing vegetation type, topography and aerosol concentration, as well as measurements from the HALO



airplane, are employed to study the total rainfall, rain rate, cloud droplet and raindrop sensitivities to these environmental and geographical characteristics.

### 3.2.1    Rainfall $D_m$ as a Function of Rain Rate for Polluted and Clean Cases

The impact of the aerosol number concentration on the cloud microphysical properties needs to be analysed for the different seasons. Particle concentrations using the CPC, at the surface, for the dry and wet seasons are very different. For instance, the 33rd and 66th percentiles are 673 cm-3 and 1377 cm-3 for the wet season and are 1954 cm-3 and 3392 cm-3 for the dry season, respectively. The dry season has nearly three times more aerosols than the wet season, which can have strong implications for

cloud and rainfall formation. However, as shown in Fig. 2, the thermodynamic characteristics are also very different, and the differences cannot be explained only by the aerosol concentration difference. To evaluate the aerosol impact on cloud processes, the rainfall mean mass-weighted diameter is evaluated for different population of particle concentrations for each season. However, the comparison was possible only for the wet season because the rainfall events in the dry season for the cases with particle concentrations exceeding 66th percentile were rare. During the dry season, the upper one-third of the

population of aerosol concentrations is characterized mostly by drier days. Figure 7 shows the $D_m$ during the wet season for background (particle concentration less than 33rd percentile) and polluted (particle concentration more than the 66th percentile) conditions as a function of rain rate. This calculation requires two different instruments to be co-located. Therefore, the population of each 5 minutes continuous rainfall events, in each rain rate class was drastically reduced. Consequently, the differences between the two air quality populations in each rain rate bin were not significant at 95%. Even if the differences

between the two average values for background and polluted cases are not significant, the physical results are coherent. For small rain rates, which are more often associated with warm cloud processes, the cases with background aerosol concentrations have a larger $D_m$ because there are fewer CCN and lower cloud droplet concentration, resulting in large raindrops. However, for higher rain rates, typically associated with deep convection, the $D_m$ is higher for polluted clouds, demonstrating the major effect of convective invigoration discussed in the preceding section. Rosenfeld and Ulbrich (2003) using satellite data

estimated the raindrop size distributions for continental and maritime Amazon clouds (LBA). Clouds over the continent produces greater concentrations of large drops and smaller concentrations of small drops. They suggested this behavior is caused by the aerosol effects on the precipitation forming processes.

### 3.2.2    Rainfall Sensitivity to Surface Cover

There is a very complex diurnal cycle over the Amazonas. Burleyson et al. (2016) used 15 years of satellite data to show a heterogeneous spatial distribution of convection that results from numerous local effects of the rivers and vegetation cover. Saraiva et al. (2016), using radar data, also found the Amazonas diurnal cycle to be regionally varying. In addition to the natural geographical effect, there is rainfall modulation by anthropogenic-induced changes in vegetation and by the presence





of large cities. Laurent et al. (2003) showed the differing cloud cover as a function of the deforestation pattern. Lin et al. (2010), among several other studies, discussed the urban heat island effect on the climate. To understand how vegetation cover influences the precipitation characteristics, two approaches were studied: one using statistical radar data and another using a special HALO mission specifically aimed at this topic.

For the statistical approach, the SIPAM S-band radar was employed to compute the rain rate for the dry and wet seasons, as described in Section 2. The surface cover was obtained from the digital Terra-Class classification (Almeida et al., 2008). This database presents 15 classes of vegetation such as forest; hydrology; urban areas; and several classes of deforestation including, clean and dirty pasture, deforested areas, and exposed soil, among others. These classes were regrouped in four specific classes: forest (covering 76.9% of the studied region), hydrography (16.3%), non-forest (6.2%), and urban area (0.5%). These two sets

of data were combined to evaluate the different rain rates for each surface cover type. Terra-Class has 300 m resolution and was interpolated to the radar grid considering the most frequent surface type class.

Figure 8 shows the rain rate box plot statistics for each surface type in the wet and dry seasons. This analysis does not consider the different thermodynamic or dynamics conditions or the cloud life cycle. It only considers the overall statistics among the different surface types. The different behaviours are consequence of different physical processes in which will be discussed in

this section.

For the wet season, the rain rate statistics vary little among the different surface cover classes. However, for the dry season, one can note differences such as a higher amount of rainfall over the urban areas and smaller amounts over non-forested regions. As expected, in general, the dry season rain rate is higher than that in the wet season. The median and the tail of the distribution are larger over urban areas and smaller over deforested areas. The variation between the non-forest and urban rain

rate for the dry season is approximately 25%. Although the statistical population of radar pixels in each class are very different and the urban area represents only 0.5% of the area, the difference is very consistent. The few differences during the wet season are expected as the rainfall, as described above, has a strong component of stratiform clouds typical of a monsoon rainfall regime. In this type of regime, the large-scale dynamic forcing is very strong, and the surface type has little impact. However, during the dry season, rainfall events largely depend on local forcing and surface latent heat flux. Manaus, as an urban centre,

is characterized by a strong heat island (Souza and Alvala, 2012) that creates convergence fed by moisture from the surrounding forest. The non-forest area has less available latent heating than the forest which might contribute to lower rain rate. We should clarify that these are the results for the rain rate and not for the total rainfall amount. The total rainfall amount is larger over the forest and hydrology areas (not shown).

Fisch et al. (2004) discussed the differences in the boundary layer between forest and pasture. They show, for the Amazonas

dry season, that the height of the fully developed convective boundary layer over forest is approximately 1100 m and approximately 50% higher over pasture. During the wet season, the forest and pasture have nearly the same boundary layer height (approximately 1000 m). Does the different thermodynamic behaviour during the dry season result in different cloud microphysical properties and, consequently, different radiative forcing? This is a very important question because climate change simulations in Amazonas need to correctly reproduce the cloud processes in each of these surface covers. The results



presented above consider the rain rate and the final effect of the surface on the cloud. Several physical processes play an important role in the generation of the different rain rates between different surface types, such as the different boundary layers described above. The effect of the surface type on the cloud droplet distribution on the shallow convection is not well elaborated. One of the ACRIDICON-CHUVA HALO missions was especially designed to investigate this feature. One could

consider that the cloud processes for different surface covers could be evaluated with a combination of several different flights from a statistical point of view as a function of the surface cover. However, airplane flights are limited to a few hours and the flights measure the cloud for different meteorological conditions, heights, thermodynamic conditions, and aerosol loadings that, as already stated in the set of papers published using the GoAmazon data, have a strong impact on cloud microphysical properties. Therefore, a specific flight mission was designed to evaluate this matter. The flight AC17, completed on 27

September 2014, was a mission looking at the cloud contrast between forest and pasture surfaces. The objectives were to observe and compare clouds macrophysical and microphysical properties over both forest and pasture areas at comparable environmental conditions. As the flight legs occurred at the same fixed level, the thermodynamics and aerosol concentrations were nearly the same due to the short flight time and path, only 40 km, in each region. The two flight-leg paths are shown in Figure 9. The cloud profiling was carried out over the 80 km legs (red lines in Fig. 9), which are forested, transition, and

pasture areas. The flight plan for the cloud profiling legs was designed for a fixed altitudes level in each leg to profile clouds during the trajectories. The flight level was selected as a function of the cloud development at the local time for each flight leg. Leg 1 occurred at 1500 UTC (11:00 LST) and leg 2 at 1700 UTC (13:00 LST). The height of the flights employed in this study were 1500 m, 1900 m and 2500 m. In leg 2, the cloud base was higher and the clouds were measured at 1900 m and 2500 m. In Figure 9, the GOES visible image shows the increase of cloud cover from the first flight to the second and the

typical shallow cumulus clouds measured.

The analyses were performed using 350 seconds of measurements in each region and at each flight level. In this way, the beginning of the flight over the forest, the boundary between forest-deforested region and the final flight path over the deforested region were discarded, and only the centres of the forest and deforested region were evaluated. Figure 10a shows a scatter plot of the cloud droplet number concentration and liquid water content for flight leg 1 over the forest and pasture at

around 1500 m, 1900 m and 2500 m. At a fixed cloud droplet number concentration, clouds over the forest area have more liquid water than clouds over pasture at the same height. This means cloud droplets developing over the rain forests are larger than those evolving over pastures. Therefore, the cloud droplet size distribution is different between the two regions, with forest producing larger cloud droplets.

Figure 11 shows the cumulative histogram of the CCN number concentration ($N_{CCN}$) for the two legs over the forest and non-

forest regions at around 1500, 1900 and 2500m. The decrease of $N_{CCN}$ with the altitude for boundary layer aerosol agrees to the other flights during ACRIDICON-CHUVA as shown in Andreae et al. (2017). The magnitude of the $N_{CCN}$ is typical for polluted regions in Brazil, means in between strong biomass burning events and forest regions far off from biomass burning events. For Leg 1, $N_{CCN}$ distribution is nearly identical for 1500 and also in 1900 m height at both forest/non-forest regions, however for 2500 m, the distributions are quite difference, while non-forest has larger $N_{CCN}$. For Leg 2, the difference between



the two surface types is larger and non-forest regions present higher $N_{CCN}$ than over forest for all altitudes. In general, the difference between forest and deforest regions increase with the altitude and deforested regions always have more $N_{CCN}$. This result indicates a strong vertical mixing of the aerosol particles in the lower atmosphere and influences of the aerosol of deforest regions on forest regions. Two additional factors should be considered: leg 1 occurs at around 11:00 LST, therefore the

convective boundary layer was not fully developed and at 2500 m the measurements represent probably the residual boundary layer from the previous day. Leg 2, occurred later at around 13:00 LST, when the boundary layer was deeper and well mixed. Another important factor to be considered is the regional wind direction from east, therefore, for leg 1 the air over the forest was the air advected from the deforested region, for leg2, a north-south transect flight, the air was advected from the homogenous forest at the easterly side.

The cloud microphysics differences between forest and deforested clouds are probably related to this differences in the $N_{CCN}$ distributions. One can note that some clouds over pastures have the same amount of liquid water as clouds over forests; although pasture clouds have higher cloud droplets number concentration. This could result from several processes, such as larger vertical motion induced by the higher sensible heating and/or by the large aerosol concentration over pasture and/or the high-water availability over forest.

Figures 10b and 10c show the $D_m$ and cloud droplet number concentration as function of the vertical motion. It is clear in these figures that the larger cloud droplet diameter population over the forest and the larger cloud droplets number concentration increasing with the vertical velocity (updraft and downdraft), but there is not a clear relationship between vertical velocity and $D_m$. The vertical velocity increases supersaturation but does not appear to modulate the droplet size. Cecchini et al. (2017a) also found different flights to have a small correlation between the vertical motion and droplet size. Nevertheless, the cloud

droplet concentration is nearly linearly related to the vertical motion. The stronger the updrafts are, the higher the number concentration is. This relationship does not appear to show differences between forest and pasture areas.

### 3.2.3    Rainfall Sensitivity to Topography

The topography database employed in this study was the digital elevation data from NASA Shuttle Radar Topographic Mission with resolution of 90 m at the equator (Jarvis et al., 2008). As the data population decreases logarithmically as the elevation increases, the classes were binned in log intervals as follows: from 0 to 15 m, from 15 to 40 m, from 40 to 83 m and from 83 to 157 m. Using the S-band radar data, a statistical box plot was built for each topography class (Figure 12). For the wet season, one can note that the statistics are nearly the same for all topography classes. The highest topography class shows nearly the

same median but a slightly smaller tail distribution. However, for the dry season, significant differences are found among the topography classes. The higher the topography is, the higher is the tail of the rain rate distribution. During the dry season, convective inhibition is higher, as shown in Fig. 2, and the cloud formation requires some kind of forcing to overcome this inhibition and take advantage of the higher CAPE available during this season. Topography is one of the forcing, even if the differences are only a few hundred metres.





## 4. Summary and Conclusions

The Amazonas climate entails distinctive and complex interactions among a multitude of different physical processes, resulting in one of the most important rainfall production systems on Earth. The interannual variability is high and in recent years, records of driest and wettest years have been observed. El Niño, La Niña, and the Atlantic Ocean temperature are some of the interannual features affecting the total rainfall. Alternatively, there are many synoptic conditions responsible for the large-scale rainfall mechanisms. These are the ingredients necessary to generate large amounts of rainfall in Amazonas, normally organized in mesoscale convective systems. The cloud and precipitation systems differ in the wet and dry seasons. The total amount of rainfall in the wet season is 4.2 times larger than in the dry season, but the dry season rain rate is approximately 22% higher than that in the wet season. The wet season has a smaller CAPE, CINE, vertical wind shear, and cloud base height and a larger amount of integrated water vapor than the dry season. The wet season typically has monsoon-type rainfall, while during the dry season convection is organized at a smaller scale than in the wet season. The typical cloud cluster in Amazonas (wet and dry season) has an effective radius of approximately 75 km and a 1.5-hour life cycle. The rain cells inside these cloud clusters have an average radius of approximately 7.5 km and an 0.6-hour life cycle. The seasonality also modulates the size distribution. The wet season has more small and large cloud clusters and rain cells, typical of isolated cumuliform convection and monsoon rainfall cloud organization. The dry season has proportionally more cloud clusters and rain cells of approximately 40 km and 10 km radii, respectively, favouring cloud organization reduced in size but larger than that of isolated convection. The differences between the two seasons are also observed in the cloud microphysics. During the dry season, rainfall drops are larger than during the wet season, for the convective clouds probably due to enhanced ice processes. For stratiform clouds, larger drops are also observed, but are not statistically significantly different.

The cloud hydrometeor vertical profile signature was evaluated for the first time in Amazonas. X-band dual polarization radar data was used to provide dual polarization CFAD variables for the dry and wet seasons. As expected, there are differences between the dual polarization statistical distributions between the seasons. The wet season cloud type has a typical bright band at approximately 4 km altitude. Conversely, the dry season has a stronger reflectivity below and above the melting layer, characteristic of the liquid water and ice profiles of stronger convective clouds. The ZDR profile in the dry season indicates more vertically oriented ice, while KDP presented larger positive values in the mixed phase, consistent with more frequent and stronger updrafts. The correlation coefficient indicated more homogenous clouds above 8-km height during the dry season. This indicates a more homogenous ice layer above this level during the dry season (well-defined glaciation layer).

The evaluation of the effect of aerosol number concentration on the raindrop size distribution shows an interesting feature. Due to the small statistical sample of rainfall events during the dry season for different ranges of aerosol loadings, it was only possible to evaluate the aerosol effect during the wet season. Clean cases show larger raindrops for lower rain rates, and polluted cases show larger raindrops for a higher rain rates. For a rain rate less than 8 mm h$^{-1}$, the typical warm rain and less organized convective rainfall events, the clean cases have a more straightforward interpretation based on the small number of





CCN and consequently larger droplets. However, when convection becomes deeper with larger rain rates, the polluted cases seem to invigorate convection, as suggested by Rosenfeld et al. 2008.

General statistics for the surface type impacting the rain rate had significant results only for the dry season. The wet season had no different rain rates for different surface types. For the dry season, urban regions had the highest rain rate, and deforested regions had the lowest. This is probably related to the Manaus heat island effect, with moisture provided by the surrounding forest area and the smaller latent heating fluxes in large deforested areas, respectively. Nearly simultaneous cloud properties were measured by the HALO airplane over forest and deforested areas. The specific flight design to evaluate the microphysical differences in shallow convective cloud formation provided unprecedented data to study this difference with nearly the same synoptic and environmental conditions. Nearly simultaneous flight legs at the same height, in a short patch of only 40 km allowed us to compare the cloud processes over different surfaces types. As a result, it was observed that clouds over forest have larger cloud droplets and in general the vertical velocity is well correlated with the cloud droplet number concentration. However, it is uncorrelated with the cloud droplet mean mass-weighted diameter.

Finally, the topography impact on the rain rate was evaluated. There was no difference in the rain rate during the wet season for different topography classes. For the dry season, there was a clear increase in the rain rate as the topography increased. This was probably related to the topography forcing required to overcome the large CINE and take advantage of the large CAPE available during this season.

The results presented indicate that the GoAmazon data set brings new insights into the process of cloud and rainfall formation in the Amazon, and to complexities that need further research. There is very high potential impact of the whole data set into modelling of aerosol, cloud, and landscape features in tropical scenarios. Nevertheless, there are several fruitful areas for potential future research to complete the full picture of the cloud processes in Amazonas. For instance, the detailed microphysical description as function of the two pattern of convection, the cumuliform and deep convection. The changes in the microphysical properties and mixed phase are some of the unknown behaviours of cloud processes. How these processes change as function of the season, cloud life cycle, aerosol loading, topography are some examples that need to have further studies to improve the cloud modelling over continental tropical regions. Another potential area for future detailed studies is the implications and solutions for GCMs that may not resolve such subtle variations in topography and is very important for to trigger convection during the dry season.

*Acknowledgements:* We thank all participants in the GoAmazon2014/5 and ACRIDICON-CHUVA for the great cooperation, which made this study possible. We acknowledge FAPESP (São Paulo Research Foundation) Projects 2009/15235-8, 2014/08615-7 and 2015/14497-0. The work was conducted under scientific licenses 001030/2012-4, 001262/2012-2 and 00254/2013-9 of the Brazilian National Council for Scientific and Technological Development (CNPq). Institutional support was provided by the Central Office of the Large-Scale Biosphere Atmosphere Experiment in Amazonia (LBA), the National Institute of Amazonian Research (INPA), the National Institute for Space Research (INPE), and



Amazonas State University (UEA), and the Brazil Space Agency (AEB). We acknowledge the support of the ACRIDICON-CHUVA campaign by the Max Planck Society, the German Aerospace Center (DLR), and the German Science Foundation (Deutsche Forschungsgemeinschaft, DFG) within the DFG Priority Program SPP 1294. We also acknowledge the Atmospheric Radiation Measurement (ARM) Climate Research Facility, a user facility of the U.S. DOE, Office of Science, sponsored by

5      the Office of Biological and Environmental Research, and support from the ASR program.

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

10    .



# Figure captions

**Figure 1:**. Rain rate histogram for wet and dry seasons computed using the disdrometer in T3 site. Wet and dry seasons covered the periods between January to March and August to October, respectively, for the years 2014 and 2015. Mean values for rain rate (RR) and total rainfall (R) are given in the legend.

**Figure 2:** Box Plot of the monthly statistics of the a) convective available potential energy (CAPE), b) convective inhibition energy (CINE), c) precipitable water vapor (PWV), d) lifting condensation level (LCL) and e) bulk shear. Calculations are for 2014 using the T3 radiosondes at 00, 06, 12 and 18 UTC. The box represents the 25% to 75% populations and the line inside the box shows the median value, and the circles are the outliers.

**Figure 3:** Rain cell (based on the SIPAM S-band radar) and cloud cluster (based on GOES-13 IR brightness temperature [$B_T$]) size distributions, on the left and right side respectively, between wet and dry seasons and the difference between dry and wet season distributions (in black, right axis)

**Figure 4:** Raindrop mean mass-weighted diameter ($D_m$) as a function of the rain rate (from radar S-band) for wet and dry seasons for a) convective and b) stratiform rain events. The arrows on the x-axis indicate that the averages are different, based on the Student's t-statistic at 95% confidence. The box represents the 25% to 75% populations and the line inside the box shows the median value, and the circles are the outliers.

**Figure 5:** X-band radar reflectivity contoured frequency by altitude diagram (CFAD) for the dry and wet seasons from the X-band radar.

**Figure 6:** ZDR, KDP and horizontal-vertical correlation contoured frequency by altitude diagram (CFAD) for the dry and wet seasons from the X-band radar.

**Figure 7:** Rainfall mean mass-weighted diameter as a function of the rain rate (from radar S-band) during the wet season for clean (CPC smaller than the 33[rd] percentile) and polluted (CPC larger than the 66[th] percentile) over pasture. The box represents the 25% to 75% populations, and the line inside the box shows the median value, and the circles are the outliers.

**Figure 8:** Rain rate (from radar S-band) box plot statistics for the wet and dry seasons for different surface cover classes. The arrows on the x-axis indicate that the two (wet and dry seasons) averages are significantly different using the t-student at 95% confidence. The box represents the 25% to 75% populations, and the line inside the box shows the median value.

**Figure 9:** AC17 flight paths over forested and deforested regions. The flight leg is shown in red (source from Google Earth), the upper panel is leg#1, and the lower panel is leg#2. The dot in the flight leg corresponds to 56°57'W, 4°13'S for leg#1 and 55°17'W, 5°53´S for leg#2. Visible GOES-13 images at the time of the flights are shown on the right side for each flight leg.

**Figure 10:** a) Cloud droplet concentration as a function of the liquid water content; b) cloud mean mass-weighted diameter as a function of the vertical velocity; and c) cloud droplet concentration as a function of the vertical velocity for forest and pasture at different heights.

**Figure 11:** Figure 11: CCN number concentration ($N_{CCN}$) Cumulative histogram for leg1 (a) and leg2 (b) for AC17 flight paths over forest and non-forest regions for different flight heights.



**Figure 12:** Rain rate box plot statistics for the wet and dry seasons for different topography classes. The arrows on the x-axis indicate that the two (wet and dry seasons) averages are significantly different using the Student's t-statistic at 95% confidence. The box represents the 25% to 75% populations, and the line inside the box shows the median value. The line represents the remaining population.





## Figures

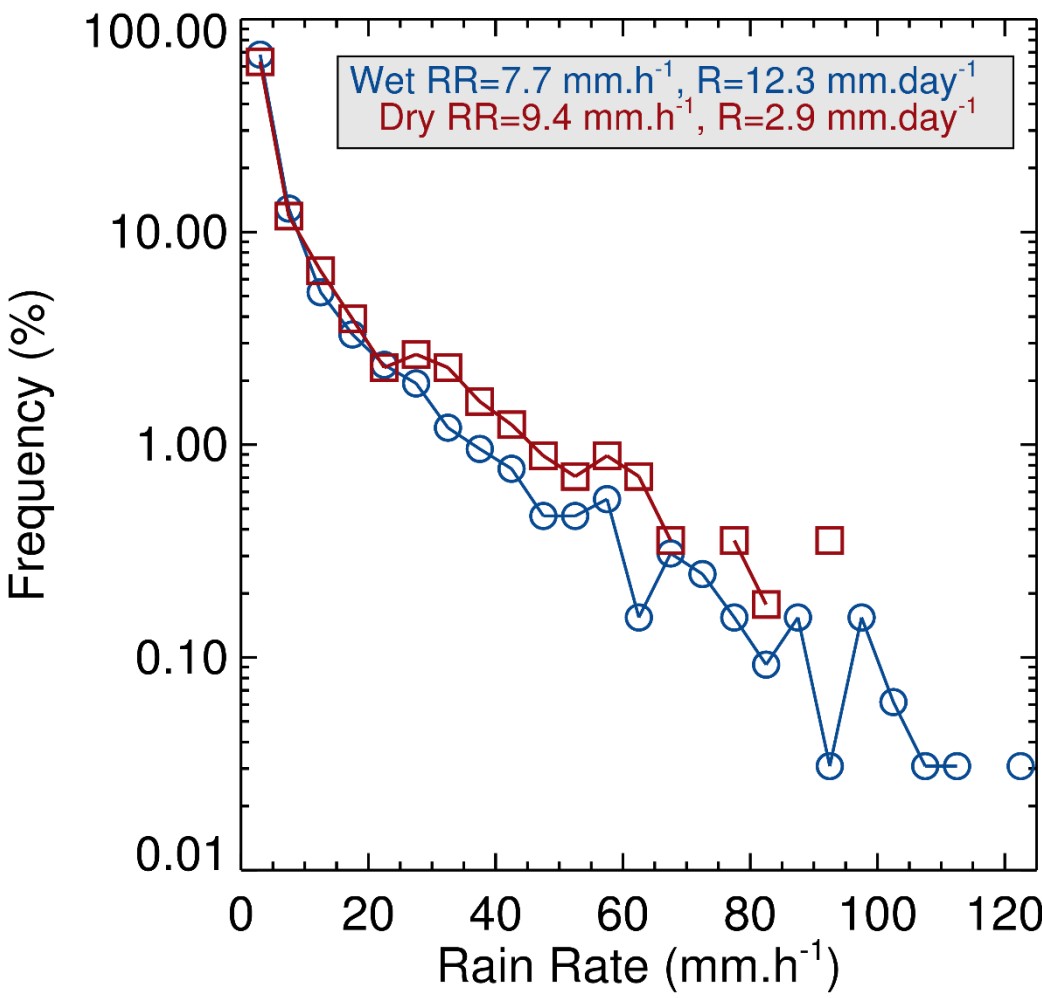

Figure 1. Rain rate histogram for wet and dry seasons computed using the disdrometer in T3 site. Wet and dry seasons covered
the periods between January to March and August to October, respectively, for the years 2014 and 2015. Mean values for rain
rate (RR) and total rainfall (R) are given in the legend.







Figure 2: Box Plot of the monthly statistics of the a) convective available potential energy (CAPE), b) convective inhibition energy (CINE), c) precipitable water vapor (PWV), d) lifting condensation level (LCL) and e) bulk shear. Calculations are for 2014 using the T3 radiosondes at 00, 06, 12 and 18 UTC. The box represents the 25% to 75% populations and the line inside the box shows the median value, and the circles are the outliers.





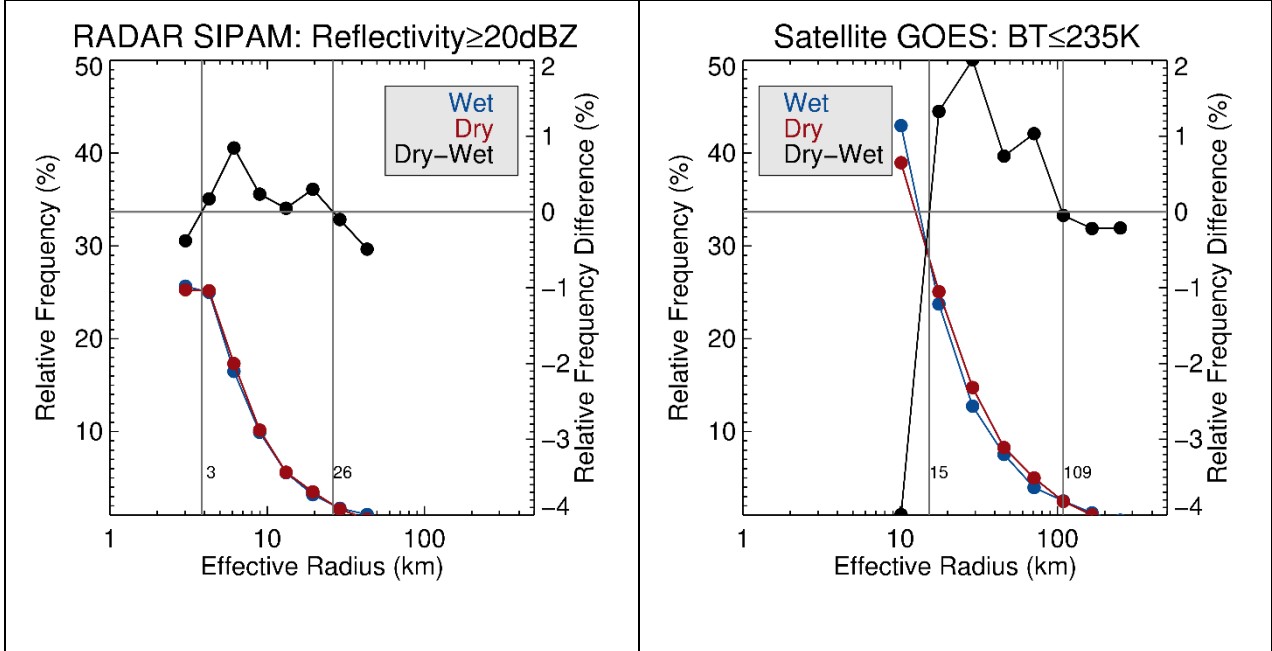

Figure 3: Rain cell (based on the SIPAM S-band radar) and cloud cluster (based on GOES-13 IR brightness temperature [$B_T$]) size distributions between wet and dry seasons and the difference between dry and wet season distributions (in black, right

5    axis)



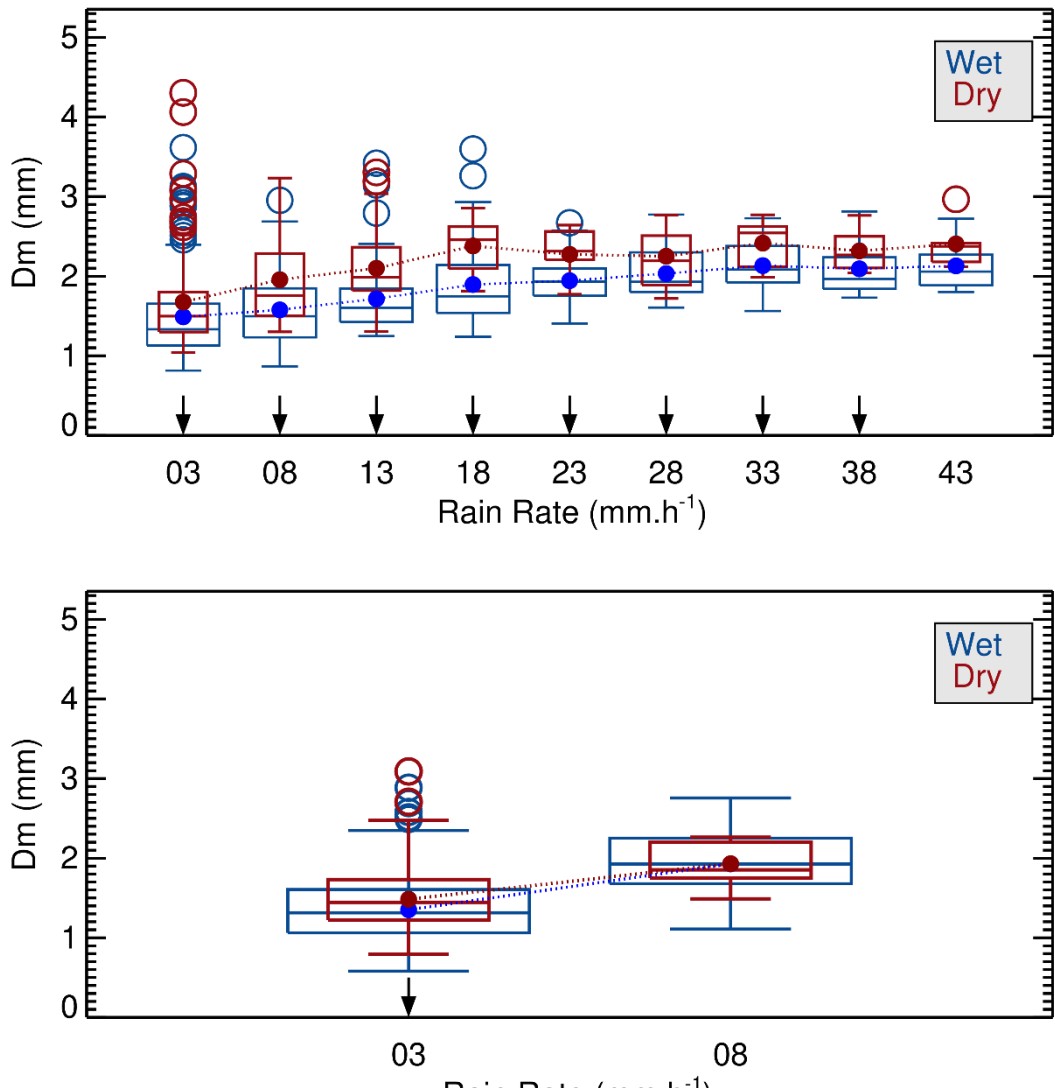

Figure 4: Figure 4: Raindrop mean mass-weighted diameter ($D_m$) as a function of the rain rate (from radar S-band) for wet and dry seasons for a) convective and b) stratiform rain events. The arrows on the x-axis indicate that the averages are different, based on the Student's t-statistic at 95% confidence. The box represents the 25% to 75% populations and the line inside the box shows the median value, and the circles are the outliers.





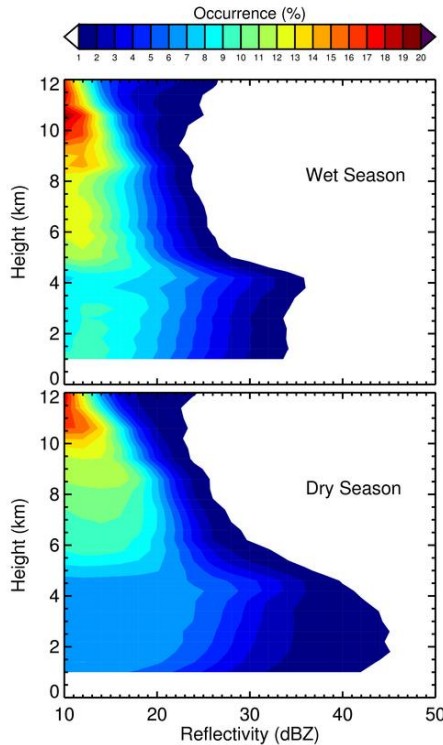

Figure 5: X-band radar reflectivity contoured frequency by altitude diagram (CFAD) for the dry and wet seasons from the X-
5   band radar.





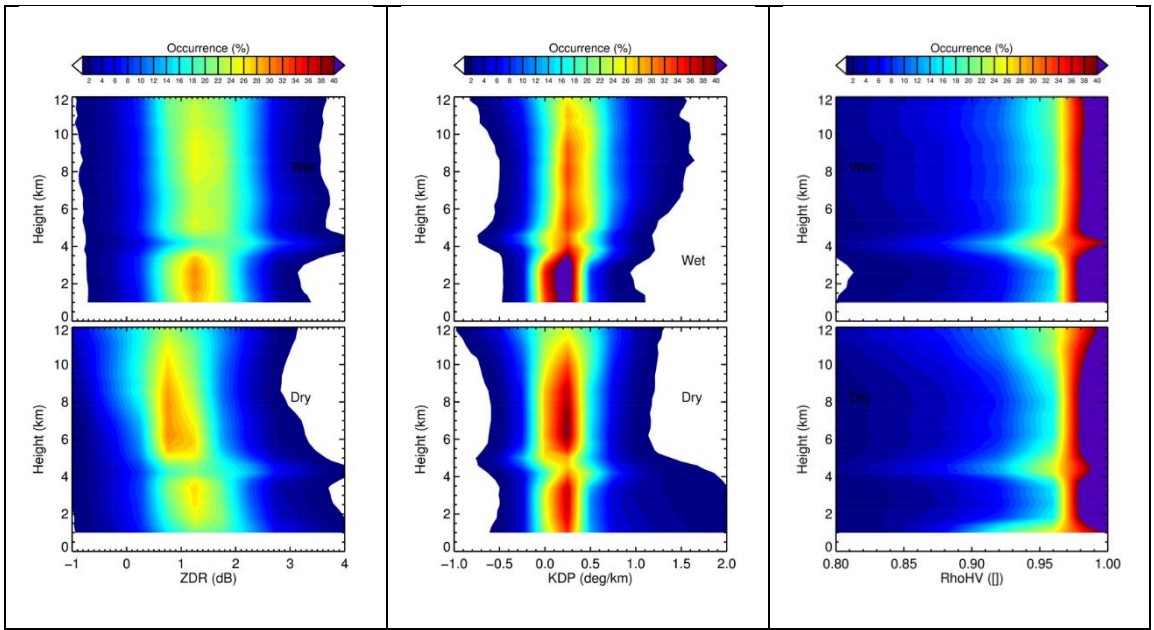

Figure 6: ZDR, KDP and horizontal-vertical correlation contoured frequency by altitude diagram (CFAD) for the dry and wet

5     seasons from the X-band radar.





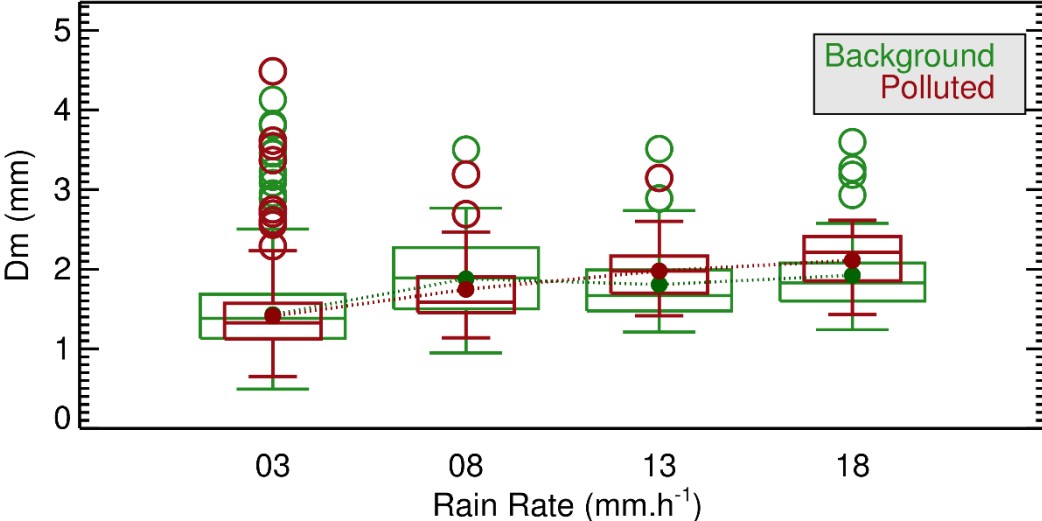

Figure 7: Rainfall mean mass-weighted diameter as a function of the rain rate (from radar S-band) during the wet season for clean (CPC smaller than the 33rd percentile) and polluted (CPC larger than the 66th percentile) over pasture. The box represents the 25% to 75% populations, and the line inside the box shows the median value, and the circles are the outliers.





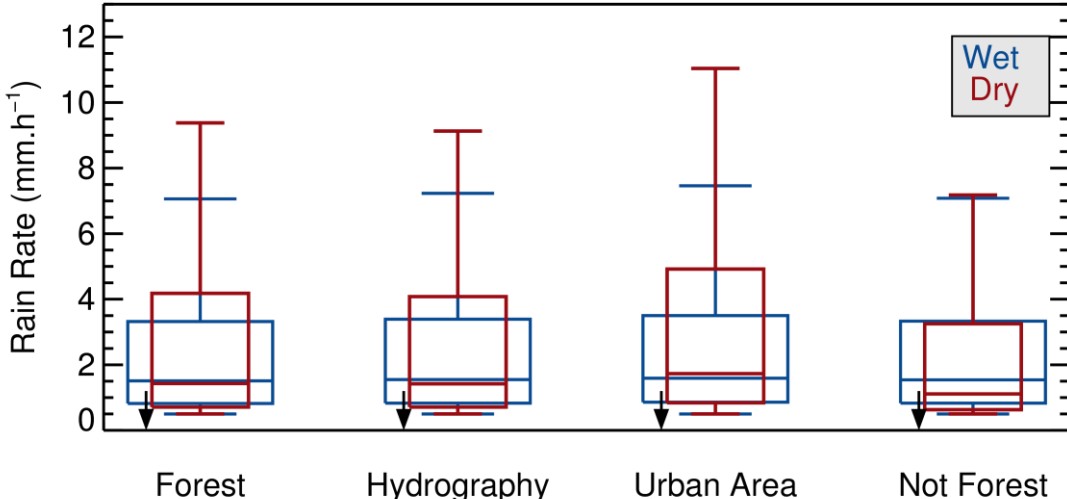

Figure 8: Rain rate (from radar S-band) box plot statistics for the wet and dry seasons for different surface cover classes.
5 The arrows on the x-axis indicate that the two (wet and dry seasons) averages are significantly different using the t-student at 95% confidence. The box represents the 25% to 75% populations, and the line inside the box shows the median value



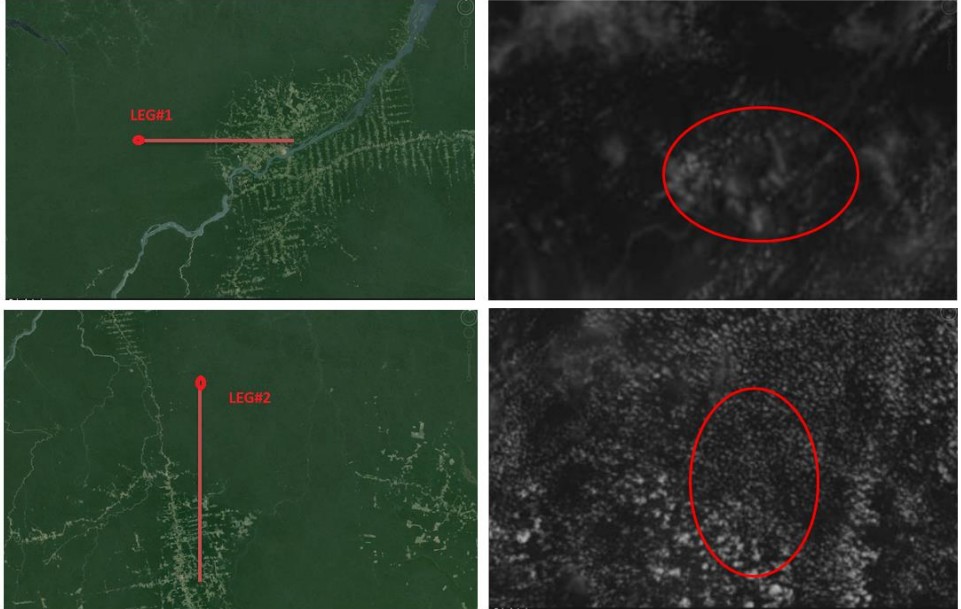

Figure 9: AC17 flight paths over forested and deforested regions. The flight leg is shown in red (source from Google Earth), the upper panel is leg#1, and the lower panel is leg#2. The dot in the flight leg corresponds to 56°57'W, 4°13'S for leg#1 and 55°17'W, 5°53´S for leg#2. Visible GOES-13 images at the time of the flights are shown on the right side for each flight leg.

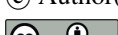



Figure 10: a) Cloud droplet concentration as a function of the liquid water content; b) cloud mean mass-weighted diameter as

a function of the vertical velocity; and c) cloud droplet concentration as a function of the vertical velocity for forest and pasture

at different heights



5    Figure 11: CCN number concentration ($N_{CCN}$) Cumulative histogram for leg1 (a) and leg2 (b) for AC17 flight paths over forest and non-forest regions for different flight heights.



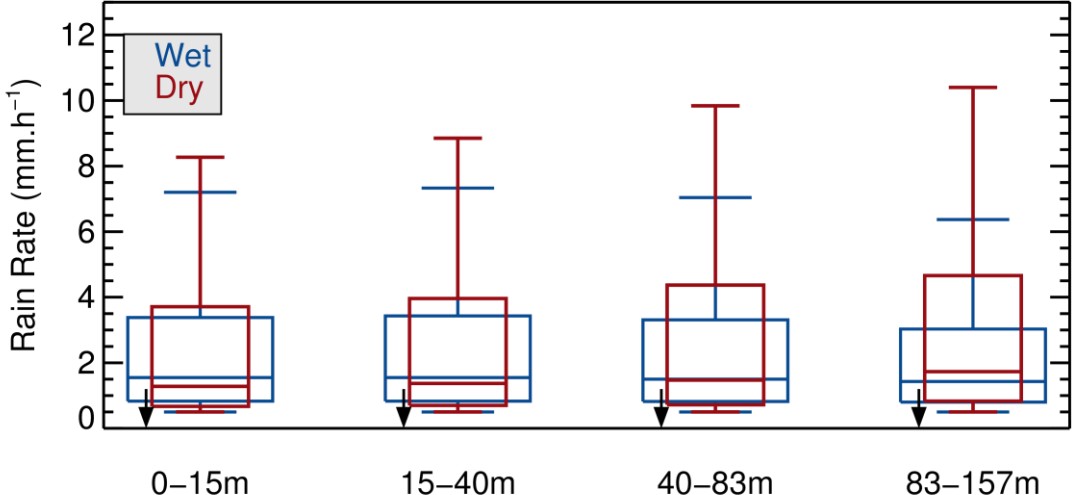

Figure 12: Rain rate box plot statistics for the wet and dry seasons for different topography classes. The arrows on the x-axis indicate that the two (wet and dry seasons) averages are significantly different using the Student's t-statistic at 95% confidence. The box represents the 25% to 75% populations, and the line inside the box shows the median value. The line represents the remaining population.