# Peer review of "Overview: Precipitation Characteristics and Sensitivities to Environmental Conditions during GoAmazon2014/5 and ACRIDICON-CHUVA"

_Atmospheric Chemistry and Physics, 2017_

## Referee Comment (RC1) · Y. Zhuang (Referee) · 5 Dec 2017

This study utilized field campaign data collected during GoAmazon 2014/5 and CHUVA-ACRIDICON, as well as satellite and S-band radar data, to give an overview of precipitation characteristic and corresponding thermodynamic conditions, and analyze the relationship between precipitation and several environmental conditions, including aerosol loading, land surface, etc. Contrast between the wet and dry season for these characteristics and relationship were emphatically discussed. Although there are numerous previous studies about the convection and precipitation in the Amazon, this is

the first paper which summarizes such complex features about the precipitation and its seasonality in the Central Amazon using multiple comprehensive datasets. Overall, I found this work to be well-written and scientifically sound, and results in this work will aid to further understanding of cloud and precipitation systems in Amazon and potentially provide implications for modeling groups to improve GCM parameterization. I recommend this manuscript to be published after some minor revisions.

Specific Comments:

1. Page 3, Line 10: Could the authors provide reference for this statement? Additionally, I think it would be also helpful to add monthly rainfall in Figure 2 since SIPAM product is available in 2014 and 2015 whole year. In ECMWF reanalysis data and S-band radar rain rate derived by (Zhuang et al. 2017, JGRA), Sep is even drier than July and August.

2. Page 3, Line 15: It would be helpful to describe how CAPE was calculated either here or in the method section. Specifically, what is the initial condition of the parcel (surface based, mixed layer average, . . .)? The choice of initial parcel could affect the CAPE value very significantly, possibly the seasonality as well.

3. Page 7, Line 12: Is there a reason for only using 2014 wet-season disdrometer data to determine the Z-R relationship? Can the authors further speculate how much this approximation that the wet and dry seasons have same DSD could affect results? Such as Figure 1 and Figure 4, does the approximation make the difference between wet and dry season smaller or larger?

4. Page 9, Line 21-22: I feel that usage of "rain rate" and "rainfall" could be a little confusing here. By "This is the reason why the wet season has a maximum rain rate", I think the authors actually mean the average daily rain rate but not the rain rate used in Line 15-16 for rain event. I feel it's better to explicitly specify the average period and use something like "rain rate for precipitation event", or just use symbol RR and R to discriminate them. Also check Line 34 in abstract and description in Figure 1.

[Figure]

5. Page 9, Line 30-33: Similar conclusions about atmospheric instability and cloud fraction variations between wet and dry season were also discussed in some previous studies such as Zhuang et al. 2017.

6. Page 9, Line 31: Definition of bulk shear is not given. Is it surface to 6km bulk shear?

7. Page 12, Line 24-26: Is the comparison "clearer distinction" made between the dry and wet season or between 8km and above 8km in the dry season? Also, in conclusion section at Page 17, Line 28-29, "... more homogenous clouds above 8-km ...", does this contradict with "... higher correlation at approximately 8km" here? It seems to me the frequency of RhoHV<0.97 is higher above 8km, and that of RhoHV>0.97 is lower above 8km. Doesn't this mean the average RhoHV is smaller and the cloud becomes less homogenous above 8km?

8. Comparison between Figure 6c and 6f shows dry season has larger frequency in high RhoHV range (larger purple area) above melting layer and below 8 km, which means RhoHV is smaller and the cloud becomes less homogenous above 8km.

9. Page 14, Line 19-22: Discussions here are not very clear. What does "the variation ... is 25%" mean? What "difference" is "very consistent"?

10. Page 16, Line 2: "difference ... increase with the altitude ...". I don't find this statement to be true for Figure 11b.

11. Page 16, Line 20: Please provide statistical test for the linear relationship in Figure 10, especially 10c. In addition, solid rectangles and circles look very similar in Figure 10bc. Maybe use another marker such as "x" in Figure 10a.

12. Page 16, Line 30-31: Firstly, although dry season seems to have a stronger linear dependency between rainfall and elevation than the wet season, they still look very similar. Is this difference significance tested between all adjacent elevation groups? Similarly, it would be helpful to indicate if the difference passes the significance test between different surface types for a single season in Figure 8. Secondly, the conclusions

here about the dependency of dry season rain rate on topography seem to valid at first. However, is this result independent from those in section 3.2.2 about surface type? If so, the authors need to indicate there is no specific relationship between surface type and topography. I also suggest adding a figure to show surface type and contoured elevation of the studied area.

13. Page 18, Line 21: Could the authors provide reference to the related studies?

14. Quality of some figures need to be improved. Specifically, e.g., sub-figures were not properly labelled, such as Figure 3-6 & 9; black lines around the figure should be removed, such as Figure 3&6; Figure 2, Maybe more details about the box plot can be given either in text, figure caption, or both. e.g. how is "outlier" defined and how to determine the length of whiskers?; text "wet" and "dry" are not all visible inside the Figure 6; Figure 10a is in different size with 10bc; some texts were not shown as subscripts, such as Nccn and Dm; etc.

15. I'm not sure how to interpret the unit (%) of occurrence frequency in Figure 5&6. If the CFAD was constructed the same way as (Yuter and Houze, 1995, Part II, MWR), shouldn't the unit be, for example, "% km-1 dBZ-1" for Figure 6ac.

Typos and Grammar Issues includes, but is not limited to:

Page 1, Line 28: "This is study" –> "This study"

Line 30: "instruments systems" –> "instrument systems"

Line 32: "have carefully been" –> "have been carefully"

Line 35: "While" cannot be used to start the sentence here

Page 2, Line 1: "as well" –> "as well as", "among" –> "between"

Line 2: "analyse" –> "analyzed"

Line 3: "is" –> "was"

Line 7: "observe" –> "observed", "dependence on" –> "dependence of"

Line 10: "cloud droplets number" –> "cloud droplet number"

Line 10-11: "revealed", "exhibit" check tense consistency

Line 20: "sea -level" –> "sea level"

Page 3, Line 10: "Amazonas, For" –> "Amazonas. For"

Page 4, Line 20: "During" –> "during"

Page 6, Line 20: "present" –> "presents"

Line 25: "Section two" –> "Section 2"

Page 9, Line 16: "differences, the" –> "differences. The"

Page 10, Line 6: "Cloud Clusters and Rain Cells-Size Distribution" –> "Size Distribution of Cloud Clusters and Rain Cells"

Line 27: "diameter" –> "Diameter"

Page 11, Line 4: "present" –> "presented"

Line 17: "function" –> " functions"

Page 14, Line 21: "few differences" –> "smaller differences"?

Page 15, Line 34: "difference" –> "different"

Page 17, Line 25: remove "Conversely, "

Page 30, Figure 1: Label for x-axis should be "mmïĆđh-1" not "mm.h-1". Also check other figures. Use "Sep" instead of "Sept".

Page 32, Figure 3: "distributions between wet and dry seasons and the difference between dry . . ." –> "distributions during the wet and dry seasons and the difference between the dry . . ."

Page 33, Figure 4: "t-statistic" –> "t-test"

Page 36, Figure 7: "radar S-band" –> "S-band radar"

Page 37, Figure 8: "t-student" –> "Student's t-test"

———————————————————

---

## Referee Comment (RC2) · Anonymous Referee #2 · 1 Mar 2018

This paper uses satellite and in situ data from two recent field campaigns to provide an overview of precipitation characteristics in the central Amazon, and their sensitivity to environmental conditions including time of year (wet vs dry season), aerosol concentrations, land-surface type and topography. The paper describes the complex interactions between different processes in the region, particularly through their impact on cloud microphysics, in a way which is only made possible by these new measurements. While the broad scope of the paper means that each aspect cannot be explored in a lot of detail, it still provides interesting results while also showcasing the potential of these

new datasets for further work. The paper is well organized and mostly well written (some grammar issues aside), and I recommend it for publication after addressing the following fairly minor comments.

General comments

Language:

While the paper is perfectly readable and understandable, there are minor grammar errors throughout – these do add up to quite a large number, which is why I haven't listed them below. I would encourage a thorough proofread by a native speaker.

Introduction:

This is quite long (about a quarter of the whole paper), although it is very comprehensive. I don't think it's a major issue, but worth pointing out.

Methods:

Given one of the aims of the paper is to showcase a new dataset, it is really lacking in contextual information, including where exactly the whole experiment is taking place. If the instruments are all exactly collocated simply the latitude/longitude might be ok, but I would strongly encourage you to include a map somewhere, showing the location of the instruments (particularly if placed at different locations), as well as the flight paths. This would also allow you to add some much need context. I would suggest including land surface type, topography and maybe potentially mean winds/some other climatological data. State more precisely in the abstract where the experiment is taking place (i.e. not just 'Central Amazon Basin', but ' in the vicinity of Manaus' or something like that).

Land surface results (3.2.2/3.2.3): while these results are interesting as a very general overview, I think it is difficult to draw particularly strong conclusions from them. Firstly, I'm not sure in Figure 8 there really is 95% confidence that the results are different; the test assumes all data points are independent, which will clearly not be the case. The
most obvious example is the 'urban area', which accounts for only 0.5% of points – presumably these points are all clustered together, and likely to be highly autocorrelated. Even if the differences were significant, potential cofounding factors are not considered at all by the authors. For example, topography and land surface type could be correlated in some way, in which case it wouldn't be clear which factor was really driving the differences. Finally, the explanation of physical mechanisms is sometimes inconsistent. In particular, p14, L24-26 states that the urban heat island over Manaus will drive convergence and enhanced rainfall, while reduced latent heating will decrease rainfall over non-forest. These statements are interchangeable – cities have reduced latent heating, and the non-forest will be warmer, so why do they have opposite feedbacks?

Minor comments

P9, L18-19: "Figure 1 clearly reveals..." Looking at figure 1 it looks to me like the only bin where the wet season is higher is the lowest one (and marginally, the second), which represent RR < 5.

P11, L20: "This result suggests....the wet season" I don't quite understand this sentence.

P13, L14-15: "During the dry season...mostly by drier days". Might be useful to add a short comment as to why? Presumably this is because biomass burning is more likely to occur on dry days? More broadly, some comments on what the different sources are for the aerosol you measure would be useful.

P13, L19: What was the significance level? I think it's fine to discuss the results even if the significance is below 95% if they are still physically consistent, but there is still a difference between, for example, 80% significance and no correlation whatsoever.

P15, L21-28: if clouds were at different heights over forest and non-forest, could your fixed-height measurements simply be a reflection of what part of the cloud you were measuring, instead of the clouds having different microphysical properties over different surfaces?

Figure 1: it would be nice to have error bars (these could replace the squares and circles). I would only refer to the 'T3 site' in the caption if its location is defined in the text (not just with a reference).

Figure 6: It would be helpful to state in the caption roughly what ZDR, KDP and horizontal-vertical correlation refer to physically (e.g. ice orientation for ZDR).
* * *

---

## Author Response (AR1)

**Response to Anonymous Referees**

**"Overview: Precipitation Characteristics and Sensitivities to Environmental Conditions during GoAmazon2014/5 and ACRIDICON-CHUVA" by Luiz A. T. Machado et al.**

Dear Editor,

The authors would like to thank the two reviewers for their helpful comments and suggestions. We have responded to each reviewer comment and incorporated the reviewer suggestions into the manuscript.

The individual reviewer comments and responses are included below, with author comments in **bold** and reviewer comments in *italics*.

**Response to Referee #1 – Dr. Y. Zhuang.**

We would like to thank Dr. Y. Zhuang for the valuable comments (*italic*). We will improve the manuscript based on your suggestions. Please find a point-by-point response (**bold**) and proposed changes to the manuscript below.

*This study utilized field campaign data collected during GoAmazon 2014/5 and CHUVA-ACRIDICON, as well as satellite and S-band radar data, to give an overview of precipitation characteristic and corresponding thermodynamic conditions, and analyze the relationship between precipitation and several environmental conditions, including aerosol loading, land surface, etc. Contrast between the wet and dry season for these characteristics and relationship were emphatically discussed. Although there are numerous previous studies about the convection and precipitation in the Amazon, this is the first paper which summarizes such complex features about the precipitation and its seasonality in the Central Amazon using multiple comprehensive datasets. Overall, I found this work to be well-written and scientifically sound, and results in this work will aid to further understanding of cloud and precipitation systems in Amazon and potentially provide implications for modeling groups to improve GCM parameterization. I recommend this manuscript to be published after some minor revisions.*

**Thank you for your comments. The manuscript was improved based on your recommendations.**

*Specific Comments:*

1. *Page 3, Line 10: Could the authors provide reference for this statement? Additionally, I think it would be also helpful to add monthly rainfall in Figure 2 since SIPAM product is available in 2014 and 2015 whole year. In ECMWF reanalysis data and S band radar rain rate derived by (Zhuang et al. 2017, JGRA), Sep is even drier than July and August.*

**The reference is Machado et. al. (2004) DOI 10.1007/s00704-004-0044-9. In this case, we are referring to Manaus (data were obtained from rain gauges). You are correct that September is drier than July and that August is the driest month. September was added to the text. We added an additional Figure (Fig. 2F) to show the 2014 monthly mean rainfall and rainfall rates. We believe that it is more appropriate to use the rain gauge data in this instance. The SIPAM S-band radar data are discussed in item 3, and 2014 was chosen to reflect the same period shown in the other panels of Figure 2.**

2. *Page 3, Line 15: It would be helpful to describe how CAPE was calculated either here or in the method section. Specifically, what is the initial condition of the parcel (surface based, mixed layer average, . . .)? The choice of initial parcel could affect the CAPE value very significantly, possibly the seasonality as well.*

**We added the thermodynamic calculation procedures to the methodology based on your suggestion. This procedure relied on surface calculations. The Williams et al.**

**paper also relied on surface data to compute CAPE (this information was also included in the text).**

> 3. *Page 7, Line 12: Is there a reason for only using 2014 wet-season disdrometer data to determine the Z-R relationship? Can the authors further speculate how much this approximation that the wet and dry seasons have same DSD could affect results? Such as Figure 1 and Figure 4, does the approximation make the difference between wet and dry season smaller or larger?*

**There was a mistake in the legends for Figures 4 and 7; the rainfall data were collected using a disdrometer, as explained in the text. The effect of the radar S-band rainfall estimation is only considered in Figures 8 and 12. As these figures are presented by vegetation and topography class for each season, the Z-R relationship has little effect on the conclusions. Therefore, the variation between the classes rather than the absolute values should be considered. We used the Joss disdrometer and the period with the best data (i.e., the wet season with the J-W) to create the Z-R relationship. A sentence was added to the text discussing the possible implications of this method on the total rainfall measurements.**

**The Z-R relationship is more sensitive to the way one filters out disdrometer data than to the intrinsic difference between the wet and dry Z-R relationship. Thus, this relationship was considered by ARM-DOE as more appropriate (Courtney Schumacher, data mentor). The Joss disdrometer was only used during the wet season by the researchers because they were considering a more continuous and larger sample than previously collected. The researchers filtered out all points with less than 100 droplets per minute. This method resulted in a more conservative relationship that is closer to Marshall Palmer. However, if a filter of 10 drops per minute is considered, the Z-R relationship differs more substantially than can be explained by the wet-dry seasonal differences. Figure 1 shows some of the Z-R relationships used to test the ARM Z-R relationship. IOP2-JW is the Joss adjusted relationship during the dry season (filter 10 drops/minute); IOP1-JW is the Joss-adjusted relationship during the wet season (filter 10 drops/minute); and Aaron is the relationship used in the manuscript and in the ARM database (for wet Joss but filtered using 100 drops per minute). The Marshall Palmer (MP) and NEXRAD were also considered. Note that differences between how the disdrometer data are filtered are larger than the differences between dry and wet seasons when using the same filter. We added a discussion within the text explaining why the S-band data were not considered as an absolute value of rainfall.**

[Figure]

4. *Page 9, Line 21-22: I feel that usage of "rain rate" and "rainfall" could be a little confusing here. By "This is the reason why the wet season has a maximum rain rate", I think the authors actually mean the average daily rain rate but not the rain rate used in Line 15-16 for rain event. I feel it's better to explicitly specify the average period and use something like "rain rate for precipitation event", or just use symbol RR and R to discriminate them. Also check Line 34 in abstract and description in Figure 1.*

**We were trying to explain the exception (the largest RR occurred during the wet season – only one case) and in doing so created confusion, as you noted. We rewrote this part to clarify our meaning and have taken your other suggestions into consideration.**

5. *Page 9, Line 30-33: Similar conclusions about atmospheric instability and cloud fraction variations between wet and dry season were also discussed in some previous studies such as Zhuang et al. 2017.*

**The reference was added to the text.**

6. *Page 9, Line 31: Definition of bulk shear is not given. Is it surface to 6km bulk shear?*

**The definitions of all the thermodynamic-dynamic parameters were added to the methodology. The bulk shear is the difference between the surface-500 m and surface-6000 m average wind speed.**

7. *Page 12, Line 24-26: Is the comparison "clearer distinction" made between the dry and wet season or between 8km and above 8km in the dry season? Also, in conclusion section at Page 17, Line 28-29, ". . . more homogenous clouds above 8-km . . .", does this contradict with ". . . higher correlation at approximately*

*8km" here? It seems to me the frequency of RhoHV<0.97 is higher above 8km, and that of RhoHV>0.97 is lower above 8km. Doesn't this mean the average RhoHV is smaller and the cloud becomes less homogenous above 8km?*

**The clearest distinction appears below and above 8 km. This was clarified in the text. You are correct that the sentence was not well written, which resulted in an incorrect interpretation. The text was changed to "During the dry season, there appears to be a clearer distinction between the mixed phase and the glaciation phase above 8 km. The wet season correlation coefficient is more homogenous with height inside the cloud."**

8. *Comparison between Figure 6c and 6f shows dry season has larger frequency in high RhoHV range (larger purple area) above melting layer and below 8 km, which means RhoHV is smaller and the cloud becomes less homogenous above 8km.*

**We agree and have addressed this item as discussed in item 7. The paragraph has been rewritten. We hope the new text is clearer and more accurate.**

9. *Page 14, Line 19-22: Discussions here are not very clear. What does "the variation . . . is 25%" mean? What "difference" is "very consistent"?*

**Yes; this sentence requires clarification. We intended to refer to the difference between the RR median values of each surface type. In addition, the data are not consistent but are significant. This was changed in the text.**

10. *Page 16, Line 2: "difference . . . increase with the altitude . . .". I don't find this statement to be true for Figure 11b.*

**You are correct that the results are only clear for leg 1. This correction was made in the text. The time of leg 1 compared with leg 2 (later, when the convective boundary layer was fully developed) resulted in this difference, which has been clarified in the text.**

11. *Page 16, Line 20: Please provide statistical test for the linear relationship in Figure 10, especially 10c. In addition, solid rectangles and circles look very similar in Figure 10bc. Maybe use another marker such as "x" in Figure 10a.*

**The statistical information (correlation) was revised in the text. As the correlation is only around 0.6, we changed the sentence from nearly linearly related to more linearly related than the vertical velocity.**

12. *Page 16, Line 30-31: Firstly, although dry season seems to have a stronger linear dependency between rainfall and elevation than the wet season, they still look very similar. Is this difference significance tested between all adjacent elevation groups? Similarly, it would be helpful to indicate if the difference passes the significance test between different surface types for a single season in Figure 8. Secondly, the conclusions here about the dependency of dry season*

*rain rate on topography seem to valid at first. However, is this result independent from those in section 3.2.2 about surface type? If so, the authors need to indicate there is no specific relationship between surface type and topography. I also suggest adding a figure to show surface type and contoured elevation of the studied area.*

**We agree that it is more appropriate to test significant differences among the classes than between the seasons. We have tested the differences among the classes, and all differences during the dry season were significant. However, we continued our analysis of this difference using the T-student parametric test. We were curious why all classes passed the test and identified important discussion points related to parametric significance tests in very large sample sizes. These comparisons were done for several days at the pixel level, so the sample set is very large. All tests generally pass under this condition, and there is no statistical way to use the test with such a large sample set. By using box plots, all basic statistics are shown. Therefore, we decided to eliminate the T-student test for vegetation and topography. This test was only valid when we tested T3 site-specific data, which resulted in a much smaller sample set. We eliminated the arrows in Figure 9 (new, formerly 8) and 13 (new, formerly 12).**

*13. Page 18, Line 21: Could the authors provide reference to the related studies?*

**A reference was added to the text.**

*14. Quality of some figures need to be improved. Specifically, e.g., sub-figures were not properly labelled, such as Figure 3-6 & 9; black lines around the figure should be removed, such as Figure 3&6; Figure 2, Maybe more details about the box plot can be given either in text, figure caption, or both. e.g. how is "outlier" defined and how to determine the length of whiskers?; text "wet" and "dry" are not all visible inside the Figure 6; Figure 10a is in different size with 10bc; some texts were not shown as subscripts, such as Nccn and Dm; etc.*

**The sub-figures are now labeled The frame was a placeholder in the manuscript. The figures will be provided individually, and the Journal will organize according to their standards. Details about the box plots were added to the text. The "Wet" and "Dry" labels in Figure 6 are now visible. The size of Figure 10 was changed.**

*15. I'm not sure how to interpret the unit (%) of occurrence frequency in Figure 5&6. If the CFAD was constructed the same way as (Yuter and Houze, 1995, Part II, MWR), shouldn't the unit be, for example, "% km-1 dBZ-1" for Figure 6ac.*

**The CFAD was constructed as follows: each CFAD consists of a PDF of reflectivity at each height multiplied by 100 so that the values are presented as percentages. The CFAD calculation used 2 dBZ-bin and 0.4 km-bin intervals. After the first paper (Yuter and Houze, 1995), the others papers that use CFAD rely on this explanation. We added this sentence to the legend and the intervals.**

*Typos and Grammar Issues includes, but is not limited to:*

*Page 1, Line 28: "This is study" —> "This study"*

**Changed as recommended.**

*Line 30: "instruments systems" –> "instrument systems"*

**Changed as recommended.**

*Line 32: "have carefully been" –> "have been carefully"*

**Changed as recommended.**

*Line 35: "While" cannot be used to start the sentence here*

**Changed as recommended.**

*Page 2, Line 1: "as well" –> "as well as", "among" –> "between"*

**Changed as recommended.**

*Line 2: "analyse" –> "analyzed"*

**Changed as recommended.**

*Line 3: "is" –> "was"*

**Changed as recommended.**

*Line 7: "observe" –> "observed", "dependence on" –> "dependence of"*

**Changed as recommended.**

*Line 10: "cloud droplets number" –> "cloud droplet number"*

**Changed as recommended.**

*Line 10-11: "revealed", "exhibit" check tense consistency*

**Changed as recommended.**

*Line 20: "sea -level" –> "sea level"*

**Changed as recommended.**

*Page 3, Line 10: "Amazonas, For" –> "Amazonas. For"*

**Changed as recommended.**

*Page 4, Line 20: "During" –> "during"*

**Changed as recommended.**

*Page 6, Line 20: "present" –> "presents"*

**Giangrande et al., so present is correct.**

*Line 25: "Section two" –> "Section 2"*

**Changed as recommended.**

*Page 9, Line 16: "differences, the" –> "differences. The"*

**Changed as recommended.**

*Page 10, Line 6: "Cloud Clusters and Rain Cells-Size Distribution" –> "Size Distribution of Cloud Clusters and Rain Cells"*

**Changed as recommended.**

*Line 27: "diameter" –> "Diameter"*

**Changed as recommended.**

*Page 11, Line 4: "present" –> "presented"*

**Changed as recommended.**

*Line 17: "function" –> " functions"*

**Changed as recommended.**

*Page 14, Line 21: "few differences" –> "smaller differences"?*

**Changed to smaller differences.**

Page 15, Line 34: "difference" –> "different"

**Changed as recommended.**

Page 17, Line 25: remove "Conversely, "

**Changed as recommended.**

Page 30, Figure 1: Label for x-axis should be "mm ïˊCd¯h-1" not "mm.h-1". Also check

other figures. Use "Sep" instead of "Sept".

**Changed as recommended**

Page 32, Figure 3: "distributions between wet and dry seasons and the difference between dry . . ." –> "distributions during the wet and dry seasons and the difference between the dry . . ."

**Changed as recommended.**

Page 33, Figure 4: "t-statistic" –> "t-test"

**Changed as recommended.**

Page 36, Figure 7: "radar S-band" –> "S-band radar"

**This part was eliminated as explained in item 3.**

Page 37, Figure 8: "t-student" –> "Student's t-test"

**Changed as recommended.**

**Response to Anonymous Referee #2**.

We would like to thank you for your valuable comments (*italic*). We will improve the manuscript based on your suggestions. Please find a point-by-point response (**bold**) and proposed changes to the manuscript below.

*This paper uses satellite and in situ data from two recent field campaigns to provide an overview of precipitation characteristics in the central Amazon, and their sensitivity to environmental conditions including time of year (wet vs dry season), aerosol concentrations, land-surface type and topography. The paper describes the complex interactions between different processes in the region, particularly through their impact on cloud microphysics, in a way which is only made possible by these new measurements. While the broad scope of the paper means that each aspect cannot be explored in a lot of detail, it still provides interesting results while also show casing the potential of these new datasets for further work. The paper is well organized and mostly well written (some grammar issues aside), and I recommend it for publication after addressing the following fairly minor comments.*

**Thank you for your comments and suggestions. All points were addressed as described below.**

*General comments*
*Language:*
*While the paper is perfectly readable and understandable, there are minor grammar errors throughout – these do add up to quite a large number, which is why I haven't listed them below. I would encourage a thorough proofread by a native speaker.*

**The manuscript has been reviewed by American Journal Experts, an English-language editing service. We hope this has improved the grammar throughout the paper.**

Introduction:
This is quite long (about a quarter of the whole paper), although it is very comprehensive. I don't think it's a major issue, but worth pointing out.

**We agree that the Introduction is long for a conventional manuscript; however, because we are treating the introduction as an overview, we wished to describe what is known about cloud processes in the Amazon as determined from the field campaign so far. This is why we split the Introduction in two sub-sections. We hope this structure is acceptable as is.**

Methods:
Given one of the aims of the paper is to show case a new dataset, it is really lacking in contextual information, including where exactly the whole experiment is taking place. If the instruments are all exactly collocated simply the latitude/longitude might be ok, but I would strongly encourage you to include a map somewhere, showing the location of

the instruments (particularly if placed at different locations), as well as the flight paths. This would also allow you to add some much need context.

**There are two papers that discuss the details of the GoAmazon campaign: Introduction: Observations and Modeling of the Green Ocean Amazon (GoAmazon2014/5) by Martin et al. (2016) (doi:10.5194/acp-16-4785-2016) and The Green Ocean Amazon Experiment (GoAmazon2014/5) Observes Pollution Affecting Gases, Aerosols, Clouds, and Rainfall over the Rain Forest by Martin et al. (2015) (doi:10.1175/BAMS-D-15-00221.1). We understand that the manuscript should be read independently from others papers; however, both papers were included in a special issue, and the introductory paper provides all of the descriptions about the project. Martin´s Figures 1 and 2 provide the information you request. In the beginning of the methodology, we reference these specific figures in order to provide the reader with a general description of the sites and flights.**

I would suggest including land surface type, topography and maybe potentially mean winds/some other climatological data.

**We added a figure (new Figure 8) indicating the radar-covered area, the vegetation and topography relative to T3 and the SIPAM radar.**

State more precisely in the abstract where the experiment is taking place (i.e. not just 'Central Amazon Basin', but ' in the vicinity of Manaus' or something like that).

**Changed as recommended.**

Land surface results (3.2.2/3.2.3): while these results are interesting as a very general overview, I think it is difficult to draw particularly strong conclusions from them. Firstly, I'm not sure in Figure 8 there really is 95% confidence that the results are different; the test assumes all data points are independent, which will clearly not be the case. The most obvious example is the 'urban area', which accounts for only 0.5% of points – presumably these points are all clustered together, and likely to be highly autocorrelated. Even if the differences were significant, potential cofounding factors are not considered at all by the authors. For example, topography and land surface type could be correlated in some way, in which case it wouldn't be clear which factor was really driving the differences.

**We have revised the figure to illustrate the vegetation and topography distributions. The data are not correlated, and forest covers much of the area. We computed the T-student test for the vegetation and topography to identify the difference between seasons, but the best method would be to test the difference among the classes, as stated by reviewer 1.**
**We agree with reviewer 1 that it is more appropriate to test significant differences among the classes than between the seasons. We have tested the differences among classes, and all differences in the dry season were significant. Furthermore, we used the T-student parametric test. We were curious why all classes passed the test and identified important discussion points related to parametric significance tests when using very large sample sizes. These comparisons were done for several days at the**

**pixel level, so the sample set is very large. All tests generally pass under this condition, and there is no statistical way to use the test with such a large sample set. By using box plots, all basic statistics are shown. Therefore, we decided to eliminate the T-student test for vegetation and topography. This test was only valid when we tested T3 site specific data, which resulted in a much smaller sample set. We eliminated the arrows in Figure 9 (new, formerly 8) and 13 (new, formerly 12).**

Finally, the explanation of physical mechanisms is sometimes inconsistent. In particular, p14, L24-26 states that the urban heat island over Manaus will drive convergence and enhanced rainfall, while reduced latent heating will decrease rainfall over non-forest. These statements are interchangeable – cities have reduced latent heating, and the non-forest will be warmer, so why do they have opposite feedbacks?

**The answer to your question involves the scale. There is a forest around Manaus, so the city will be warmer and receive moisture convergences from the forest. However, there is little moisture convergence from large deforested regions during the dry season because these regions have no moisture source to support this process (if the deforested area is large). We attempted to explain this scale problem and discuss the cautions that need to be taken when considering these physical explanations because scale impacts our discussion. The text was revised in the vegetation section to address this concern.**

Minor comments
P9, L18-19: "Figure 1 clearly reveals. . ." Looking at figure 1 it looks to me like the only bin where the wet season is higher is the lowest one (and marginally, the second), which represent RR < 5.
**This is a logarithmic scale; however, we have deleted the word "clearly."**

P11, L20: "This result suggests. . ..the wet season" I don't quite understand this sentence.
**This sentence has been removed.**

P13, L14-15: "During the dry season. . .mostly by drier days". Might be useful to add a short comment as to why? Presumably this is because biomass burning is more likely to occur on dry days? More broadly, some comments on what the different sources are for the aerosol you measure would be useful.

**We introduced a sentence with this discussion and added a reference that describes the aerosol types.**

P13, L19: What was the significance level? I think it's fine to discuss the results even if the significance is below 95% if they are still physically consistent, but there is still a difference between, for example, 80% significance and no correlation whatsoever.

**We computed the significance (85%) and added it to the text.**

P15, L21-28: if clouds were at different heights over forest and non-forest, could your fixed-height measurements simply be a reflection of what part of the cloud you were measuring, instead of the clouds having different microphysical properties over different surfaces?

**You are suggesting that clouds over forest have a different cloud base than those over non-forest. This is true; however, we are not certain if this occurs within such a short path as those in this study (around 50 km). That said, the first leg occurred in the morning when the cloud base difference is very small. There was a cloud base flight (just below) prior to the level 1500. We have added a discussion about this possible effect on the measurements in the text.**

Figure 1: it would be nice to have error bars (these could replace the squares and circles). I would only refer to the 'T3 site' in the caption if its location is defined in the text (not just with a reference).

**Figure 1 is a frequency distribution; therefore, it is not possible to add error bars. However, we added a new Figure 2f, which shows the rain rate box plots.**

Figure 6: It would be helpful to state in the caption roughly what ZDR, KDP and horizontal-vertical correlation refer to physically (e.g. ice orientation for ZDR).

**Changed as recommended.**

[revised manuscript text omitted]
 only. T. The relative population intensity of the dry season rainfall events is more pronounced towards high RR than that of events during the wet season. This distinctive feature has important consequences for the cloud microphysical and macrophysical structures of clouds. The main reason for this difference is associated with the instability and cloud cover. Figure 2 presents monthly box plots for the of the monthly statistics of thermodynamics variables, with the lower (Q1) and upper (Q3) bounds representingare the 25% and 75 percentiles. The whiskers arewere defined by Q1 - 1.5 * IQR (lower) and Q1 + 1.5 * IQR (upper).; IQR is the interquartile range (Q3-Q1). Figure 2A shows the CAPE distribution for the wet and dry seasons in 2014. The dry season has a larger CAPE than the wet seasonone, and the frequency with which the of CAPE exceedsing 2000 J kg$^{-1}$ is higher during the dry season. The wet season has typical monsoonal rainfall, with widespread moderate rain, in contrast to the more isolated and intense rainfall events that occur during the dry season. Zhuang et al.'s (2017) study ofing shallow-to-deep convection transition in Amazonas found similar results.

This characteristic of rainfall events with where a higher RRrain rate occurs during the dry season rainfall events is explained based onby the seasonal differences in the thermodynamic parameters. Comparing these seasons, important differences can be

clearly noted in Fig.Figure 2 highlights some of these important differences. The dry season has a larger CAPE, higher CINE, less available water vapour, a higher cloud base, and higher shear than the wet season. The CAPE increases from March to September; and the largest tail distributions occur at the end of the year when humidity increases and cloud base decreases. During the dry season, only regions with strong forcing can produce convective clouds thatto use the higher CAPE and shear available to produce organized convection. Gonçalves et al. (20154) show that the higherincreased RRsrain rates (radar reflectivity values larger than 35 dBZ) during the dry season mainly occurs over the higher topographyhigher elevations in Amazonas (section 3.2.1). The higher CINE and, smaller amount of water vapour reduces the occurrence of convection, but when convection is able tocan develop, it has all the ingredients to be deeper. Machado et al. (2004) explains that the more intense convective clouds during the dry to wet season transition may result from less "competition" of surface moisture convergence to feed cumulonimbus clouds because there is a smaller number of rain cells exist. Figure 2f presents the RRrain rate statistics for 2014 and the monthly rainfall measured by rain gauge in T3.

**3.1.2    Size Distribution of Cloud Clusters and Rain Cells**

Cloud clusters and rain cell calculations were computed using thedata were derived from GOES-13 satellite images and S-band radar employing using the algorithm called Fortracc (Forecasting and Tracking Cloud Clusters see Vila et al., 2008) algorithm. A cloud cluster is defined by connected ensembles of pixels with brightness temperatures (BT), for channel the 10.5 μm channel that are, colder than 235 K, as defined by Machado et al. (1988). A rRain cells is defined as a connected ensembles of pixels in the radar 2.5 km CAPPI with reflectivity larger than 20 dBZ. Embedded in the cloud clusters, qQuite often, rain cells, which are embedded in cloud clusters, are observed when clouds start to formhave raindrops. Considering the wet and dry seasons, the typicalThe average cloud cluster size and lifespan arehave 75-km effective radius (hereafter called as radius) and a 1.5 -hours, respectively, during the wet and dry seasons in the Amazon. The typical rain cell size has a 7.5-km radius and an 0.6-hour lifespan. On average, cloud clusters are 10 times larger and have lifespans oftimes approximately three times that of their associated the rain cells. These are the average characteristics; and, there is a wide range of cloud clusters come in a wide range of sizes. Cloud clusters can have more than aexceed 300 -km in radius and have a lifespantime longer than 24 hours, while rain cells can grow to sizes occur up to approximately 60 -km in radius and lifetime of alast for a couple of hours. Figure 3 shows the dry and wet season cloud clusters and rain cell size distributions identified in this study, as well asand the differences betweenamong them. The basic size distribution is not very differentdoes not vary substantially between the two seasons because cloud cluster size distribution has a power lawfollows an exponential size distribution, as shown by Machado et al. (1992); however, certain distinctions, but some clear characteristics can be noted if the difference is computed. The wet season has more small and large rain cells and cloud clusters than the dry season. The dry season produces more rain cells in the range of a 10-km radius and cloud clusters of an approximately 40-km radius. The ratio between the cloud cluster and rain cell average radii during the wet season is much higher greater because of the larger stratiform cloud decks typical of a monsoon cloud regime. The thermodynamics of the dry season environment discussed in the preceding section favours the organization

of more compact and active convection with more intense rainfall events  but  accumulated rainfall amounts that are 4 times smaller.

**3.1.3 Mass-Weighted Mean Rainfall  Diameter for the Dry and Wet Seasons**

 Variations in cloud processes between the two Amazonian seasons were evaluated in this study in order to determine whether.  important microphysical differences between raindrops during the wet and dry seasons exist, or whether only  RRrain rate and rainfall frequency vary between seasons. These features were investigated through the deployment of disdrometers and a dual-polarimetric radar. This study focuses on rainfall and raindrops; however,  seasonal differences in cloud droplet size distributions may warrant attention as well.

For instance, the effect of  aerosol concentrations on  cloud droplets in shallow convective clouds, where aerosols generally reducing the size and increasing the concentration ofor a given liquid water content, is well known (Cecchini et al., 2016, among several other studies). However, if a polluted cloud transitions from shallow-to-deep convection, aerosols can invigorate clouds (Rosenfeld et al., 2008; Koren et al., 2012; Gonçalves et al., 2015). Giangrande et al. (2017) presented the G1 airplane cloud particle distribution measurements taken during GoAmazon2014/5 and showing the predominance of larger cloud droplets in warm clouds during  the wet season. The in situ cloud droplet data were collected for a shallow cloud population. The result is very different when  seasonal data collecteds  using disdrometers that measure raindrops at least 100 times larger (in their mean mass-weighted mean diameter (Dm)) are compared. Given that the raindrop diameter depends on RRrain rate, which varie 
[revised manuscript text omitted]